# DRPO: Efficient Reasoning via Decoupled Reward Policy Optimization

**Gang Li**[1*]     **Yan Chen**[2*]     **Ming Lin**     **Tianbao Yang**[1]
[1] Texas A&M University     [2] The University of Virginia
`gang-li@tamu.edu, jdf3nk@virginia.edu`
`linming04@gmail.com, tianbao-yang@tamu.edu`

## Abstract

Recent large reasoning models (LRMs) driven by reinforcement learning algorithms (e.g., GRPO) have achieved remarkable performance on challenging reasoning tasks. However, these models suffer from overthinking, generating unnecessarily long and redundant reasoning even for simple questions, which substantially increases computational cost and response latency. While existing methods incorporate length rewards to GRPO to promote concise reasoning, they incur significant performance degradation. We identify the root cause: when rewards for correct but long rollouts are penalized, GRPO's group-relative advantage function can assign them negative advantages, actively discouraging valid reasoning. To overcome this, we propose Decoupled Reward Policy Optimization (DRPO), a novel framework that decouples the length-based learning signal of correct rollouts from incorrect ones. DRPO ensures that reward signals for correct rollouts are normalized solely within the positive group, shielding them from interference by negative samples. The DRPO's objective is grounded in integrating an optimized positive data distribution, which maximizes length-based rewards under a KL regularization, into a discriminative objective. We derive a closed-form solution for this distribution, enabling efficient computation of the objective and its gradients using only on-policy data and importance weighting. Of independent interest, this formulation is general and can incorporate other preference rewards of positive data beyond length. Experiments on mathematical reasoning tasks demonstrate DRPO's significant superiority over six efficient reasoning baselines. Notably, with a 1.5B model, our method achieves 77% length reduction with only 1.1% performance loss on simple questions like GSM8k dataset, while the follow-up baseline sacrifices 4.3% for 68% length reduction. The code is available at `https://github.com/Optimization-AI/DRPO`

## 1 Introduction

Recently, large reasoning models (LRMs) driven by Reinforcement Learning (RL) (Guo et al., 2025; Team et al., 2025b) have demonstrated remarkable performance on complex reasoning tasks like mathematics, coding, and scientific problem-solving. Unlike conventional language models that focus on direct thoughts and solutions, LRMs improve performance by generating extended chain-of-thought paths (Wei et al., 2022), allowing them to revisit intermediate steps, correct errors, and even explore alternative reasoning paths. This approach equips LRMs with stronger reasoning abilities and has become a standard paradigm to develop models capable of solving complex tasks.

However, existing LRMs suffer from overthinking with lengthy and redundant reasoning paths. As demonstrated by Chen et al. (2024), reasoning models like DeepSeek-R1 (Guo et al., 2025) need to generate about 1,000 tokens to answer "what is the answer of 2 plus 3", while only around 10 tokens are needed for non-reasoning models. Such overly generated reasoning paths raise significant issues, leading to substantially increased computational cost and longer inference time. Numerous studies have been conducted to explore ways to eliminate redundant reasoning and improve the reasoning efficiency. A popular strategy is to introduce explicit reward shaping with length penalties in RL to

---

[*]Equal Contribution

guide the model toward concise reasoning (Arora & Zanette; Huang et al.; Xiang et al.; Aggarwal & Welleck), e.g., penalizing rewards of correct answers based on reasoning length to encourage shorter reasoning. Nevertheless, almost all existing methods fall short in preserving performance while shortening reasoning, causing substantial performance loss. This raises a key question: *how to guide RL for efficient reasoning with minimal performance drop?*

We identify the root cause underlying the insufficiency of existing RL-based efficient reasoning methods. The recent advancement of training efficient reasoning models has been largely built upon the Group Relative Policy Optimization (GRPO) framework, due to its groundbreaking performance (Guo et al., 2025). GRPO's effectiveness hinges on its group-relative advantage function, which normalizes a rollout's reward against the group average to create a learning signal that distinguishes positive from negative examples. Yet, this very strength becomes its greatest weakness when moving beyond simple correctness. We demonstrate that the framework is fundamentally ill-suited for composite rewards. Incorporating a length penalty reduces the reward for correct but long answers, often pushing their group-relative advantage below zero. Consequently, GRPO is misled into interpreting verbose correct answers as negative examples, discouraging valid reasoning and creating a significant optimization barrier (Figure 1).

An effective mechanism must not only distinguish right from wrong but also efficient right from inefficient right—assigning a strong positive signal to concise answers and a weaker positive signal to verbose ones, all while suppressing incorrect reasoning. To this end, we introduce Decoupled Reward Policy Optimization (DRPO), a novel RL framework that fundamentally rethinks how learning signals are constructed. DRPO's core innovation is the decoupling of the learning signal calculation: it normalizes rewards for a correct rollout only against other correct rollouts, completely insulating them from the negative examples that corrupt GRPO's signal. This ensures that length penalty proportionally reduces the positive signal of a long correct answer but never pushes it into negative territory, thereby achieving a more favorable trade-off between efficiency and accuracy. We formalize this intuition by deriving a generalized objective in a discriminative RL framework. This objective integrates a perturbed version of the on-policy positive data distribution, where the perturbation is designed explicitly to maximize a length-based reward. We derive a closed-form solution of the perturbed distribution, which allows us to efficiently optimize the objective without any additional data collection, using only on-policy samples via importance weighting.

Our **contributions** are four-fold:

- We diagnose a critical, previously overlooked deficiency in the widely-adopted GRPO framework: its group-relative advantage function is ill-suited for correctness–length composite rewards and actively harms learning when incentivizing efficiency.
- We propose Decoupled Reward Policy Optimization (DRPO), a new paradigm that decouples learning signals for positive and negative data. DRPO provides consistent, uncorrupted policy gradients for multi-reward optimization (e.g., correctness and length).
- We derive a rigorous formulation for DRPO by integrating a reward-maximizing, perturbed positive data distribution directly into a discriminative objective. We obtain a tractable closed-form solution to the perturbed distribution, yielding a practical algorithm requiring only on-policy data with no overhead.
- We conduct experiments to demonstrate the superiority of DRPO in training efficient reasoning models, substantially outperforming strong baselines across different model sizes and various mathematical reasoning benchmarks.

## 2 RELATED WORK

**Large Reasoning Models.** Earlier structured prompting approaches such as Chain-of-Thought (CoT) (Wei et al., 2022), Tree-of-Thought (ToT) (Yao et al., 2023), and Graph-of-Thought (GoT) (Besta et al., 2024) demonstrated the importance of decomposing complex problems into intermediate steps. However, these methods rely heavily on prompting and search heuristics, lacking a unified learning framework to optimize reasoning efficiency and robustness.

The breakthrough came with DeepSeek-R1, which revealed that large reasoning models (LRMs) trained via large-scale reinforcement learning, particularly GRPO (Shao et al., 2024), can autonomously acquire advanced reasoning behaviors such as branching, verification, and backtracking.

This success inspired numerous follow-up studies: some aimed at reproducing the effectiveness of GRPO (Wen et al., 2025; Luo et al., 2025b; He et al., 2025), while others investigated its limitations and proposed refinements to further enhance reasoning performance (Yu et al., 2025; Li et al., 2025a; Chen et al., 2025; Zheng et al., 2025). In parallel, many open-weight reasoning models have adopted GRPO, including Qwen-3 (Yang et al., 2025a), GLM-4.5 (Team et al., 2025a), K2-Think (Cheng et al., 2025), and Goedel-solver (Lin et al., 2025), among others. Despite these advances, existing LRMs often suffer from *overthinking*—producing unnecessarily long and redundant reasoning even for simple problems. This work aims to directly address this issue by training LRMs to reason both *efficiently* and *effectively*.

**Efficient Reasoning in LRMs.** To address the issue of overthinking in LRMs, a variety of techniques have been proposed (Sui et al., 2025; Yue et al., 2025), including (1) training-free methods, which shorten the reasoning paths via prompt (Aytes et al., 2025; Han et al., 2024) or manipulating the decoding process (Yang et al., 2025b; Yong et al., 2025; Wang et al., 2025a; Liu et al., 2025a; Wang et al., 2025c); (2) Supervised Fine-tuning (SFT) methods, which rely on compressed reasoning datasets for finetuning. These datasets are curated via token-level selection (Yuan et al., 2025; Xia et al., 2025; Zhuang et al., 2025), step-level selection (Xiao et al., 2025; Cui et al., 2025; Wang et al., 2025b), path-level selection (Munkhbat et al., 2025; Ghosal et al., 2025); (3) RL-based methods, which carefully design reward mechanisms to guide the model to reason efficiently (Hou et al.; Aggarwal & Welleck; Arora & Zanette; Xiang et al.; Luo et al.; Huang et al.; Yi et al., 2025; Liu et al., 2025b; Xu et al., 2025; Fang et al., 2025; Li et al., 2025b).

Among these techniques, RL-based methods have been demonstrated to be one of the most effective approaches due to their scalability and flexibility. Specifically, L1 (Aggarwal & Welleck) integrates a length constraint into the reward function to encourage higher performance while meeting the length goal specified in the prompt. Arora & Zanette employs online RL with a length penalty based on the distribution of correct answers, penalizing correct responses longer than the average while encouraging those shorter than the average. ALP (Xiang et al.) adaptively adjusts length penalties according to problem difficulty, measured by pass rate, assigning stronger penalties to high pass-rate problems to discourage overlong reasoning. HAPO (Huang et al.) keeps track of the minimum length of correct responses for each question, penalizing outputs that exceed this length while rewarding those that are shorter. Nevertheless, these methods all suffer from misleading learning signals and fall short in preserving performance while shortening reasoning, due to the limitation of their adopted relative advantage function.

**Discriminative Learning for LRMs.** Discriminative learning is a classical paradigm applied widely to traditional tasks like classifications (Bishop & Nasrabadi, 2006; Yang & Ying, 2022) and rankings (Burges et al., 2005; Cao et al., 2007). These methods follow the principle of raising scores for positive (correct) samples while lowering scores for negative (incorrect) ones. Recently, several works have explored applying the principle of discriminative learning to LRM training. For example, Li et al. (2025a) proposed a discriminative constrained RL framework with verifiable binary rewards to finetune LRMs. Lyu et al. (2025) and Bai et al. (2025) utilize discriminative loss for behavior cloning on positive samples and policy gradient on negative samples. Su et al. (2025) employs a discriminative learning approach based on positive-negative pairs for reasoning tasks. However, these methods are limited to binary accuracy rewards and don't address the challenge of overthinking in LRMs.

## 3 LIMITATION OF INCORPORATING LENGTH PENALTY INTO GRPO

**Notations.** We study the fine-tuning of a generative reasoning model $\pi_\theta$ parameterized by $\theta$. At each learning step, the previous model is denoted by $\pi_{\text{old}}$, which is responsible for generating answers to a given set of questions. For a question $q \in \Sigma^*$ (including its prompt), the output $o \in \Sigma^*$ is sampled from $\pi_{\text{old}}(\cdot|q)$, consisting of both reasoning traces and the final answer, where $\Sigma^*$ denotes the space of all sequences of tokens with arbitrary length. More concretely, $o$ is generated sequentially at the token level: $o_t \sim \pi_{\text{old}}(\cdot|q, o_{<t})$, for $t = 1, \cdots, |o|$. The correctness reward $r_c(o|q) \in \{1, 0\}$ for a given question $q$ and its corresponding answer in the output $o$ is verified by either matching the extracted answer against the ground-true answer or a formal verification tool (Guo et al., 2025; Zhang et al., 2025). Let $\pi_{\text{old}}^+(\cdot|q)$ denote the conditional distribution of outputs when the reward is one (i.e., correct answers) and $\pi_{\text{old}}^-(\cdot|q)$ denote the conditional distribution of outputs when the reward is zero (i.e., incorrect answers). Let $[\cdot]_+ = \max(\cdot, 0)$ denote the hinge function.

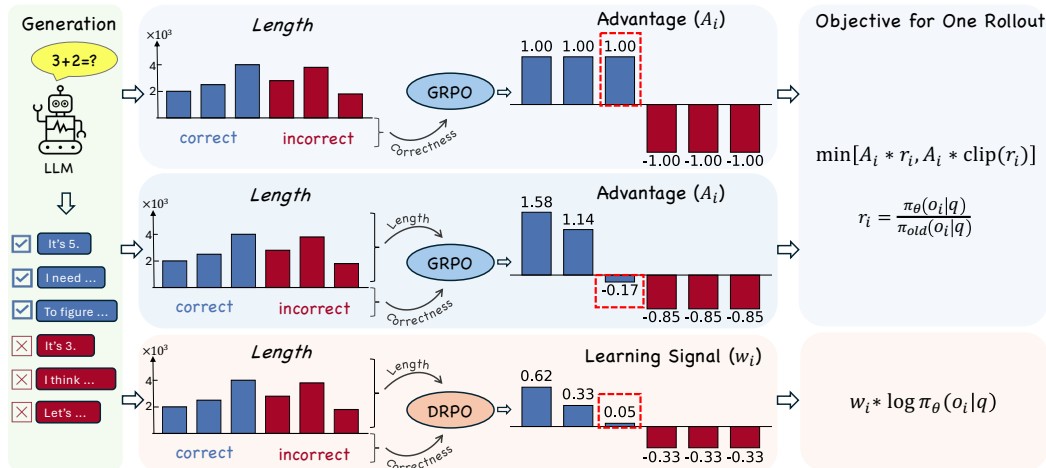

Figure 1: Illustration of the limitation of GRPO with length penalty and the benefit of our approach. Suppose $[1, 1, 1, 0, 0, 0]$ are the accuracy rewards for 6 responses, and $[0.73, 0.6, 0.2, 0, 0, 0]$ are the rewards after applying the length penalty to correct answers. Using the group-relative advantage calculation of GRPO, the advantages for the third response shift from **1** (without length penalty) to **-0.17** (with length penalty added), inadvertently penalizing the third correct response, which may substantially harm performance. In contrast, our proposed DRPO reduces the learning signal for lengthy and correct responses but never pushes them to the negative territory.

Following the success of DeepSeek-R1, Group Relative Policy Optimization (GRPO) is widely adopted in existing RL-based efficient reasoning methods to estimate the relative advantage from group rewards instead of the critic model. In the following, we illustrate the limitations of incorporating length penalty into GRPO for promoting efficient reasoning. We note that similar limitations occur to other RL methods, such as RLOO (Ahmadian et al., 2024), and other REINFORCE-based methods (Hu et al., 2025; Chu et al., 2025), which couple rewards for correct and incorrect answers to compute advantages for learning.

The GRPO objective for maximization is given by:

$$\mathcal{J}_{\text{GRPO}}(\theta) = \mathbb{E}_q \mathbb{E}_{\{o_i\}_{i=1}^G \sim \pi_{\text{old}}(\cdot|q)} \tag{1}$$

$$\left[\frac{1}{G} \sum_{i=1}^{G} \frac{1}{|o_i|} \sum_{t=1}^{|o_i|} \min\left(r_{i,t} A(o_i|q), \text{clip}(r_{i,t}, 1-\epsilon, 1+\epsilon) A(o_i|q)\right)\right] - \beta \mathbb{D}_{\text{KL}}(\pi_\theta \| \pi_{\text{ref}})$$

where $r_{i,t} = \frac{\pi_\theta(o_{i,t}|q, o_{i,<t})}{\pi_{\text{old}}(o_{i,t}|q, o_{i,<t})}$, $\pi_{\text{ref}}$ is a frozen reference model, $\mathbb{D}_{\text{KL}}(\cdot, \cdot)$ denotes the KL divergence between two distributions, and $A(o_i|q)$ denotes the relative advantage of output $o_i$, which quantifies how much better the reward of $o_i$ denoted by $r(o_i|q)$ compared to the group average reward and guides the learning direction. Specifically, $A(o_i|q)$ is computed by

$$A(o_i|q) = \frac{r(o_i|q) - \text{mean}(r(o_1|q), r(o_2|q), \cdots, r(o_G|q))}{\text{std}(r(o_1|q), r(o_2|q), \cdots, r(o_G|q))}. \tag{2}$$

Let us consider incorporating a length penalty into GRPO. Existing RL methods with length control reveal a shared principle: penalizing rewards of correct answers based on reasoning length to encourage shorter reasoning, e.g., $r(o|q) = r_c(o|q) - r_l(o|q)$, where $r_l(\cdot)$ is a length-based cost or reward function (refer to Table 2 for detailed formulations). The consequence is that the reward of a correct with long output will be shifted down relatively. Incorporating this reward into GRPO's group relative advantage calculation may bias the intended effect, misleading the learning process. Let us consider an illustrative example in Figure 1. Suppose there are six generated outputs with different lengths, whose correctness rewards are $[1, 1, 1, 0, 0, 0]$ and corresponding lengths are $[2000, 2500, 4000, 2800, 3800, 3200]$. After combining the length reward with correctness reward,

the combined reward of each answer becomes $[0.73, 0.6, 0.2, 0, 0, 0]$ [1]. So far, it looks like that all the designs work well since short correct answers have larger positive rewards and longer correct answers have smaller positive rewards while incorrect answers have zero rewards. However, when computing group relative advantage $A(o|q)$ in GRPO (i.e., Eqn. (2)), the advantage for the third correct response shifts from **1** (without length penalty) to **-0.17** (with length penalty added). This negative signal will discourage valid reasoning and create a significant optimization barrier, which may substantially harm performance.

## 4 DRPO: DECOUPLED REWARD POLICY OPTIMIZATION

The main reason of getting a negative learning signal in GRPO for a verbose correct answer is that its reshaped reward could become less than the mean reward of all samples including positive and negative ones. Our approach to avoid this issue is to decouple the rewarding of positive and negative samples so that the length rewards are only normalized within the positive group. To this end, we develop our approach based on a recent work (Li et al., 2025a), which proposes a discriminative optimization framework (DisCO) that directly increases the generative likelihood of positive answers and decrease that of negative answers. Below, we first introduce DisCO and then present our novel approach of integrating length rewards into DisCO's objective.

### 4.1 DISCRIMINATIVE CONSTRAINED POLICY OPTIMIZATION (DISCO)

DisCO was proposed to address several inherent limitations of GRPO, including difficulty bias and clipping operations. Let $s_\theta(o, q)$ be a scoring function, which measures the generative likelihood of answer $o$ given the input $q$. In this paper, we will consider $s_\theta(o, q) = \frac{1}{|o|} \sum_{t=1}^{|o|} \log \pi_\theta(o_t|q, o_{<t})$, which is effective as demonstrated in (Li et al., 2025a). The objective of DisCO is formulated as:

$$\max_\theta \quad \mathbb{E}_q \left[ \mathbb{E}_{o \sim \pi_{\text{old}}^+(\cdot|q)} s_\theta(o, q) - \tau \log \left( \mathbb{E}_{o' \sim \pi_{\text{old}}^-(\cdot|q)} \exp \left( \frac{s_\theta(o', q)}{\tau} \right) \right) \right],$$

$$s.t. \quad \mathbb{D}_{\text{KL}}(\pi_{\text{old}} || \pi_\theta) \leq \delta, \tag{3}$$

where $\delta > 0$ is a hyper-parameter. The intuition behind this formulation is straightforward: it increases the scores of positive responses $o \sim \pi_{\text{old}}^+(\cdot|q)$ while decreasing the scores of negative responses, aggregated through a log-sum-exp function. The log-sum-exp has its roots in discriminative learning, appearing in losses such as cross-entropy and contrastive loss, and naturally emphasizes hard negatives by assigning them larger learning signal. The constraint $\mathbb{D}_{\text{KL}}(\pi_{\text{old}} || \pi_\theta) \leq \delta$, inspired by TRPO (Schulman et al., 2015), is added to ensure the stability of training.

While DisCO demonstrates impressive gains in reasoning performance over GRPO, the length of its reasoning is uncontrolled, leaving the challenge of enhancing reasoning efficiency unresolved. Moreover, the above objective is derived under a binary reward setting, which does not accept flexible reward design. In the following, we discuss how to incorporate length rewards into the framework to encourage efficient reasoning.

### 4.2 DECOUPLED REWARD POLICY OPTIMIZATION

We consider a simple length reward $r_l(o) = 1 - \frac{|o|}{C}$ for any correct response $o$, where $C$ is a constant denoting maximum response length. An intuitive idea is to assign a weight to positive answers before their scores $s_\theta(o, q)$ in (3) such that a shorter answer is assigned with a larger weight than a longer answer. Below, we formalize this idea by proposing a principled objective.

Our goal is to maximize the score of correct outputs with high length rewards while penalizing those of wrong outputs regardless of their lengths. Suppose we have a distribution $P_q^*$, which specifies a distribution of correct outputs with high length rewards given a question $q$. Then, we can modify the objective in (3) as

$$\max \mathbb{E}_q \left[ \mathbb{E}_{o \sim P_q^*} s_\theta(o, q) - \tau \log \left( \mathbb{E}_{o' \sim \pi_{\text{old}}^-(\cdot|q)} \exp \left( \frac{s_\theta(o', q)}{\tau} \right) \right) \right]. \tag{4}$$

---

[1]These values are calculated with the formula proposed in (Arora & Zanette). We note that various reward combination designs in the literature lead to the same issue. Refer to Appendix A.3 for a detailed discussion.

This can be explained that if we have an off-policy data distribution $P_q^*$ of correct outputs with high length rewards, we can use its sampled data to steer the model training such that it generates correct outputs with high length rewards more likely. However, the issue is that $P_q^*$ is not readily available. Although a naive solution is to curate such data manually, as in SFT-based efficient reasoning methods, it requires substantial human effort and lacks scalability. To address this, we draw insights from reinforcement learning from human feedback (RLHF) (Bai et al., 2022; Ouyang et al., 2022; Ziegler et al., 2019) by finding an optimal policy $P_q^*$ that maximizes the length reward with a KL regularization:

$$P_q^* = \arg \max_{P \in \mathcal{P}} \mathbb{E}_{o \sim P} r_l(o) - \lambda \mathbb{D}_{\mathrm{KL}}(P, \pi_{\mathrm{old}}^+(\cdot|q)), \tag{5}$$

where $\lambda > 0$ is a regularization parameter, $\mathcal{P}$ denotes the set of all probability measures $P$ on correct data given $q$, which are absolutely continuous with respect to $\pi_{\mathrm{old}}^+(\cdot|q)$, i.e., $\pi_{\mathrm{old}}^+(o|q) = 0$ indicates $P(o) = 0$.

However, unlike RLHF that uses a LLM to learn $P^*$, we derive its closed analytical solution similar to (Rafailov et al., 2023) (See Appendix A.1 for a complete derivation):

$$P_q^*(o) = \frac{\pi_{\mathrm{old}}^+(o|q) \exp(r_l(o)/\lambda)}{\mathbb{E}_{o \sim \pi_{\mathrm{old}}^+(\cdot|q)} \exp(r_l(o)/\lambda)}. \tag{6}$$

As a result, we have

$$\mathbb{E}_{o \sim P_q^*} s_\theta(o, q) = \sum_{o \in \Sigma^*} \frac{\pi_{\mathrm{old}}^+(o|q) \exp(r_l(o)/\lambda)}{\mathbb{E}_{o \sim \pi_{\mathrm{old}}^+(\cdot|q)} \exp(r_l(o)/\lambda)} s_\theta(o, q)$$

$$= \mathbb{E}_{o \sim \pi_{\mathrm{old}}^+(o|q)} \frac{\exp(r_l(o)/\lambda)}{\mathbb{E}_{o \sim \pi_{\mathrm{old}}^+(\cdot|q)} \exp(r_l(o)/\lambda)} s_\theta(o, q).$$

Plugging this formulation back into (4), we obtain the final objective function:

$$\max \quad \mathbb{E}_q \left[ \mathbb{E}_{o \sim \pi_{\mathrm{old}}^+(\cdot|q)} \frac{\exp(r_l(o)/\lambda)}{\mathbb{E}_{o \sim \pi_{\mathrm{old}}^+(\cdot|q)} \exp(r_l(o)/\lambda)} s_\theta(o, q) - \tau \log \left( \mathbb{E}_{o' \sim \pi_{\mathrm{old}}^-(\cdot|q)} \exp \left( \frac{s_\theta(o', q)}{\tau} \right) \right) \right]$$

$$s.t. \quad \mathbb{D}_{\mathrm{KL}}(\pi_{\mathrm{old}} || \pi_\theta) \leq \delta. \tag{7}$$

It is notable that the final objective only relies on the on-policy data, and it has an explanation that each positive data is assigned with a weight $\omega(o|q) = \frac{\exp(r_l(o)/\lambda)}{\mathbb{E}_{o \sim \pi_{\mathrm{old}}^+(\cdot|q)} \exp(r_l(o)/\lambda)}$ informed by its length but normalized only within the positive data. It is notable that when $\lambda = +\infty$, then $\omega(o|q) = 1$ and the above objective reduces to that of DisCO in (3).

We solve the optimization problem (7) similarly as (Li et al., 2025a). In particular, the expectations are replaced by empirical averages and the KL divergence is estimated by using sampled data, and the constraint is handled by adding a penalty function $\beta_0 [\mathbb{D}_{\mathrm{KL}}(\pi_{\mathrm{old}} || \pi_\theta) - \delta]_+^2$ to the objective, where $\beta_0$ is a penalty constant. For completeness, we present a full algorithm for solving (7) in Algorithm 1 in the Appendix. We refer to this method as Decoupled Reward Policy Optimization (DRPO).

## 5 EXPERIMENTS

**Datasets.** We validate our method on mathematical reasoning tasks. Specifically, we train models on the DeepScaleR-Preview-Dataset (Luo et al., 2025c), which consists of approximately 40.3k question-answer pairs sourced from AIME problems from 1984 to 2023, AMC problems before 2023, Omni-MATH (Gao et al., 2024) and Still (Min et al., 2024) datasets. We evaluate all the models on math problems with different levels of difficulty, including (a) easy level: GSM8K (Cobbe et al., 2021), (b) medium level: MATH-500 (Hendrycks et al., 2021), (c) hard level: Olympiad-Bench (He et al., 2024), and (d) very hard level: AIME (aggregating 2024 and 2025). To verify the generalizability of our method to non-mathematical reasoning tasks, we also conducted experiments on K&K logic puzzle dataset (Xie et al., 2024), which is included in Appendix A.4.

**Models.** We adopt three reasoning models as our base models: DeepSeek-R1-Distill-Qwen-1.5B model, DeepSeek-R1-Distill-Qwen-7B, and DeepSeek-R1-Distill-Llama-8B, and conduct RL fine-tuning from them.

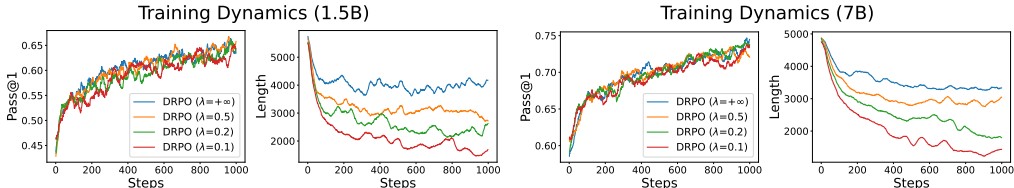

Figure 2: Training dynamics of DRPO with different regularization weights $\lambda$. The left two plots are for fine-tuning the 1.5B model, and the right two are for fine-tuning the 7B model. $\lambda = +\infty$ denotes the reference method DisCO, which does not incorporate length rewards in training.

**Baselines.** We compare our methods with six of the most recent state-of-the-art efficient reasoning methods, including (1) the method in (Arora & Zanette), which integrates length reshaped rewards into the RLOO advantage function, referred to as RLOO-LP; (2) ALP (Xiang et al.), which uses a length penalty in GRPO that is the length scaled by the solving rate of each question; (3) HAPO (Huang et al.), which penalizes the responses longer than the shortest correct answer in the history while rewarding those that are shorter; (4) L1-max (Aggarwal & Welleck), which is a reasoning language model that produces outputs satisfying a maximum length constraint given in the prompt; (5) ShorterBetter (SB) (Yi et al., 2025), which aims to match Sample Optimal Length defined as the shortest correct response among multiple generations; (6) LASER-D (Liu et al., 2025b), which employs a step length reward function based on difficulty-aware dynamic target length. We train models for methods (1)-(3) using the experimental settings described below. For methods (4)-(6), we evaluate the models provided in their original works. All the compared models were finetuned from the same base models on the DeepScaleR-Preview dataset, except L1-max, which was trained on DeepScaleR-1.5B-Preview. We summarize different reward designs of the above baselines in Table 2 in Appendix.

**Training Details.** For all the training, we employ the AdamW optimizer with a weight decay of 0.01 and set the learning rate to a constant $2\mathrm{e}^{-6}$ for 1.5B model, $1\mathrm{e}^{-6}$ for 7B model, and $5\mathrm{e}^{-7}$ for 8B model, following Li et al. (2025a). We set the batch size to 128 for each step of RL, the mini-batch size to 32 for each iteration of model update, and sample 8 responses per question for training. For RLOO-LP, we tune their weight parameter $\alpha \in \{0.05, 0.1, 0.2\}$. For ALP, we tune their penalty weight $\beta \in \{1\mathrm{e}^{-9}, 1\mathrm{e}^{-8}, 1\mathrm{e}^{-7}\}$. For HAPO, we tune their weight parameter $w \in \{0.01, 0.1, 1\}$. For the proposed method, we tune $\lambda \in \{0.5, 0.2, 0.1\}$. These parameters serve the same role that controls the tradeoff between efficiency and accuracy. For all other hyperparameters, we follow the default values from their official papers. Details are provided in Appendix A.2. The generation budget is limited to 8k tokens for both training and evaluation.

**Evaluation.** We use Pass@1 averaged over the 16 generated answers per prompt as the performance metric and use the averaged number of tokens as the length metric. For all methods, we train the model for 1000 RL steps to enable convergence and conduct evaluation every 200 steps. The models with the best pass@1 are reported, as we aim to enhance reasoning efficiency with minimal performance reduction. For models that are trained by us, we set temperature = 0.6 and top-p = 0.95, consistent with the training setup. For L1-MAX and LASER-D, we also use temperature = 0.6 and top-p = 0.95, while we use temperature = 0.9 and top-p = 0.9 for SB, following the original paper.

In addition to pass@1 and reasoning length, following (Luo et al., 2025a; Yi et al., 2025), we also adopt Accuracy Efficiency Score (AES) as a supplementary metric. AES integrates performance and reasoning length into a single measure, directly quantifying the trade-off between accuracy and computational cost. The AES is computed by:

$$\mathrm{AES} = \begin{cases} \alpha * \Delta_{\mathrm{Length}} + \beta * |\Delta_{\mathrm{Acc}}|, & \text{if } \Delta_{\mathrm{Acc}} \geq 0, \\ \alpha * \Delta_{\mathrm{Length}} - \gamma * |\Delta_{\mathrm{Acc}}|, & \text{if } \Delta_{\mathrm{Acc}} < 0. \end{cases}$$

where $\alpha, \beta, \gamma > 0$, $\Delta_{\mathrm{Length}} = \frac{\mathrm{Length_{ref}} - \mathrm{Length_{model}}}{\mathrm{Length_{ref}}}$ and $\Delta_{\mathrm{Acc}} = \frac{\mathrm{Acc_{model}} - \mathrm{Acc_{ref}}}{\mathrm{Acc_{ref}}}$, where the quantities with subscript "ref" means each method's baseline that does not consider length reward, and that with subscript "model" means the model for evaluation. In our experiments, we use the default values $\alpha = 1, \beta = 3$ same as (Luo et al., 2025a), but set $\gamma = 10$ to emphasize the importance of minimizing performance degradation.

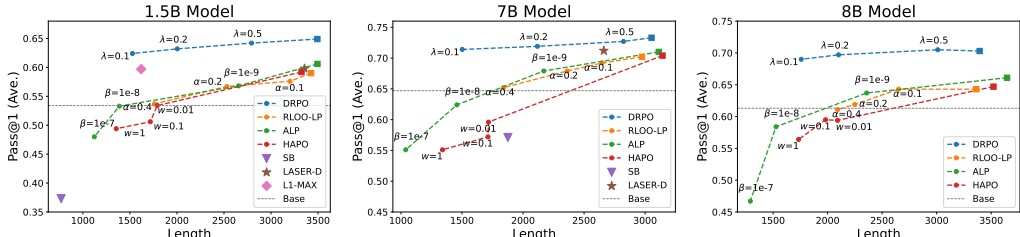

Figure 3: Comparison of performance-efficiency trade-off. Left is for fine-tuning 1.5B model, middle is for fine-tuning 7B model and right is for fine-tuning 8B model. Grey lines represent the base model performance before finetuning, with generation length of 4698 for 1.5B model, 4119 for 7B model, and 4325 for 8B model. Squares denote models trained with reference methods without length penalties, i.e., $\lambda$=+∞ (corresponding to DisCO) for DRPO, $\alpha = 0$ for RLOO-LP, $\beta = 0$ (corresponding to GRPO) for ALP, $w = 0$ for HAPO. Triangle, star, and rhombus markers represent models trained by other works.

## 5.1 VISUALIZATION OF LEARNING PROCESS

To directly verify the effectiveness of the proposed method, we present training dynamics of DRPO with different regularization weights $\lambda$ in Figure 2, where $\lambda$=+∞ corresponds to the reference method DisCO, which does not employ length reward during training. In terms of performance (first and third figures in Figure 2), we can see that DRPO with smaller $\lambda$ values exhibits marginally worse or comparable performance compared with DisCO, while the second and fourth figures show that smaller $\lambda$ values lead to substantial reductions in response length, with $\lambda = 0.1$ reducing length by over 50% relative to $\lambda$=+∞ (DisCO). These observations demonstrate the effectiveness of DRPO to achieve more concise reasoning while maintaining nearly unchanged training performance. In the following section, we will evaluate the generalization of DRPO to test datasets, compared with other strong efficient reasoning baselines.

## 5.2 COMPARISON WITH BASELINES

In this part, we evaluate the effectiveness of the proposed method on test datasets, compared with existing efficient reasoning baselines.

**Trade-off between performance and efficiency.** We present the trade-off between performance and efficiency for various methods in Figure 3, where the averaged pass@1 over four math datasets of different difficulty is reported. We observe that our proposed DRPO consistently achieves significantly better performance-efficiency trade-off than all baselines on finetuning 1.5B, 7B and 8B models, including the models trained by other work. Notably, relative to the reference learning method (square marker), the proposed DRPO on finetuning 7B model in Figure 3 (Middle) efficiently reduces reasoning length from 3053 to 1502 (51% length reduction) with only 2.6% loss of performance via varying $\lambda$ to control the tradeoff, demonstrating the effectiveness of DRPO to reduce reasoning length while preserving the reasoning capability. In contrast, all the efficient reasoning baselines suffer from severe performance degradation when the reasoning length is reduced. For example, RLOO-LP reduces reasoning length from 2975 to 1841 (38% length reduction) but incurs a 7.1% loss in performance on finetuning 7B model, with ALP showing a similar trend. Compared with DRPO, these methods trade off more performance for less reduction in length, highlighting the superiority of our method.

**Evaluating Trade-off via AES.** Additionally, we directly quantify the effectiveness of our proposed method in enhancing reasoning efficiency with minimal performance drop, using the Accuracy-Efficiency Score (AES). AES is positive when the model reduces output length while maintaining or enhancing accuracy, and negative when accuracy deteriorates. For fair comparison, reference model in AES is each method's counterpart without length reward, i.e., $\lambda$=+∞ for DRPO, $\alpha = 0$ for RLOO-LP, $\beta = 0$ for ALP, $w = 0$ for HAPO. We present the best AES score for each method in Table 1 and defer detailed results in Appendix A.6. From Table 1, we observe that almost all the baseline methods exhibit negative AES scores across 1.5B, 7B, and 8B models, indicating the inefficiency of existing methods in reducing reasoning length while preserving performance. In

Table 1: Accuracy Efficiency Score (AES) Comparison with Baselines. The best AES score for each method is presented.

|  | Method | Pass@1 | Length | AES |
|---|---|---|---|---|
| 1.5B Model | RLOO-LP | 0.567 | 2531 | -0.129 |
| | ALP | 0.606 | 3494 | -0.387 |
| | HAPO | 0.534 | 1791 | -0.519 |
| | DRPO | **0.624** | **1527** | **0.178** |
| 7B Model | RLOO-LP | 0.692 | 2649 | -0.033 |
| | ALP | 0.679 | 2170 | -0.134 |
| | HAPO | 0.596 | 1717 | -1.080 |
| | DRPO | **0.714** | **1502** | **0.249** |
| 8B Model | RLOO-LP | 0.619 | 2249 | 0.251 |
| | ALP | 0.637 | 2358 | -0.01 |
| | HAPO | 0.595 | 1981 | -0.366 |
| | DRPO | **0.690** | **1758** | **0.297** |

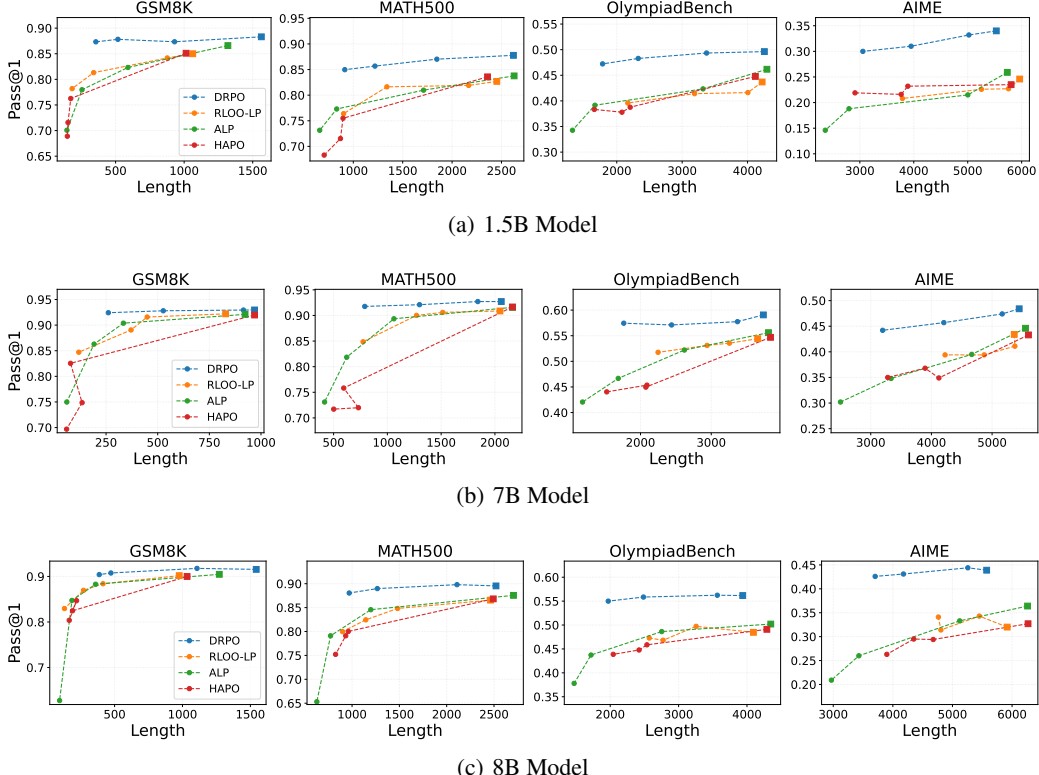

Figure 4: Performance-efficiency tradeoff on individual datasets with increasing difficulty levels from left to right. (a) is for finetuning 1.5B model, (b) is for finetuning 7B model, (c) is for finetuning 8B model. Squares denote models trained with reference methods without length penalties (i.e., $\lambda=+\infty$ (corresponding to DisCO) for DRPO, $\alpha = 0$ for RLOO-LP, $\beta = 0$ (corresponding to GRPO) for ALP, $w = 0$ for HAPO).

contrast, DRPO consistently achieves a positive AES score for finetuning 1.5B, 7B, and 8B models, highlighting its capability of improving reasoning efficiency while maintaining performance.

## 5.3 LENGTH REDUCTION FOR DIFFERENT PROBLEM DIFFICULTY

In this part, we study the impact of length reduction on questions with varying difficulties. In Figure 4, we present the performance of different methods across four math datasets of increasing difficulty, from GSM8K (easy) to AIME (very hard). As shown in Figure 4, we observe that: **(1)** on relatively easy questions like GSM8K, DRPO significantly reduces generation length from 1563 to 356 (77.2% length reduction) for 1.5B model and from 969 to 261 (73.1% length reduction) for 7B model with negligible performance drop (-1.1% for 1.5B model and -0.6% for 7B model). In contrast, all the baselines still bring dramatic performance degradation, indicated by a steep slope. **(2)** With question difficulty increasing from left to right in Figure 4, all the models exhibit longer reasoning length and introduce increasing performance drop, which is reasonable since difficult problems inherently require longer reasoning paths to solve. **(3)** When comparing Figure 4(a) and (b), with two models from the same LLM family, we can see that 7B models generally demand shorter reasoning length than 1.5B models. Besides, models with larger sizes, e.g. 7B and 8B models, are more resilient to length reduction introduced by DRPO, as evidenced by flatter slopes than 1.5B models. **(4)** Across various difficulty levels, DRPO consistently achieves a better trade-off between performance and length than all the baselines, demonstrating its effectiveness in guiding efficient reasoning.

In Appendix A.5, we conducted an ablation study on the effect of different length reward designs in DRPO, comparing (1) a linear length reward, (2) a concave length reward and (3) cosine length reward. The results are summarized in Figure 6, which shows that the linear reward provides the broadest trade-off spectrum, while the concave and cosine rewards tend to yield higher accuracy at the cost of longer reasoning lengths.

We conduct further case studies in Appendix A.7 to analyze the efficient reasoning behavior after DRPO training, compared with DisCO. It is shown that models trained by DRPO preserve existing reflection capability while effectively reducing redundant back-and-forth reasoning.

## 6 CONCLUSION AND DISCUSSIONS

In this work, we revealed a key limitation of existing RL-based efficient reasoning methods, that their reliance on group-relative advantages can mislead learning when length rewards are included. To address this, we proposed DRPO, which decouples learning signals of correct and incorrect answers so that length penalties reduce the positive signals for correct reasoning but never reverse them. Our method integrates an optimized positive data distribution under a KL regularization into a discriminative objective, for which we derived a closed-form solution, enabling efficient computation using only on-policy data with importance weighting. Experiments on math reasoning tasks demonstrate significant superiority of our approach, compared with existing efficient reasoning methods.

While DRPO is designed for efficient reasoning, its formulation is general and can be extended to incorporate other rewards on positive data beyond length, such as process rewards or other preference rewards. Extending the proposed approach to broader reasoning tasks remains an interesting direction. Another interesting direction is to incorporate mechanisms to adapt $\lambda$ to the difficulty level of questions, larger $\lambda$ for more difficult questions and smaller $\lambda$ for easier questions as implied by our results in Figure 4, which we leave for future investigation.

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

# A  APPENDIX

## A.1  DERIVING THE OPTIMUM OF THE KL-CONSTRAINED REWARD MAXIMIZATION OBJECTIVE

In this part, we derive the optimal policy $P_q^*$ that maximizes the length reward with a KL regularization:

$$\max_{P \in \mathcal{P}} \mathbb{E}_{o \sim P} r_l(o) - \lambda \mathbb{D}_{\text{KL}}(P, \pi_{\text{old}}^+(\cdot|q)), \tag{8}$$

where $\lambda > 0$ is a regularization parameter, $r_l(o)$ denotes length reward of response $o$, $\mathcal{P}$ denotes the set of all probability measures $P$ on correct data given $q$, which are absolutely continuous with respect to $\pi_{\text{old}}^+(\cdot|q)$, i.e., $\pi_{\text{old}}^+(o|q) = 0$ indicates $P(o) = 0$.

Following prior work Rafailov et al. (2023); Go et al. (2023), we have

$$\max_{P \in \mathcal{P}} \mathbb{E}_{o \sim P} r_l(o) - \lambda \mathbb{D}_{\text{KL}}(P, \pi_{\text{old}}^+(\cdot|q))$$

$$= \max_{P \in \mathcal{P}} \mathbb{E}_{o \sim P} r_l(o) - \lambda \mathbb{E}_{o \sim P} \log \frac{P(o|q)}{\pi_{\text{old}}^+(o|q)}$$

$$= \max_{P \in \mathcal{P}} \mathbb{E}_{o \sim P} \left( r_l(o) - \lambda \log \frac{P(o|q)}{\pi_{\text{old}}^+(o|q)} \right)$$

$$= \min_{P \in \mathcal{P}} \mathbb{E}_{o \sim P} \left( \log \frac{P(o|q)}{\pi_{\text{old}}^+(o|q)} - r_l(o)/\lambda \right)$$

$$= \min_{P \in \mathcal{P}} \mathbb{E}_{o \sim P} \left( \log \frac{P(o|q)}{\frac{1}{Z(q)} \pi_{\text{old}}^+(o|q) \exp(r_l(o)/\lambda)} - \log Z(q) \right) \tag{9}$$

where $Z(q) = \sum_o \pi_{\text{old}}^+(o|q) \exp(r_l(o)/\lambda)$ is a partition function, which doesn't depend on $P$.

Let's first define $\bar{P}(o|q) = \frac{1}{Z(q)} \pi_{\text{old}}^+(o|q) \exp(r_l(o)/\lambda)$. Since $\bar{P}(o|q) \geq 0$ for all $o$ and $\sum_o \bar{P}(o|q) = 1$, $\bar{P}(o|q)$ is a a valid probability distribution. Thus, we can reformulate (9) as:

$$\min_{P \in \mathcal{P}} \mathbb{E}_{o \sim P} \left( \log \frac{P(o|q)}{\frac{1}{Z(q)} \pi_{\text{old}}^+(o|q) \exp(r_l(o)/\lambda)} - \log Z(q) \right)$$

$$= \min_{P \in \mathcal{P}} \left( \mathbb{D}_{\text{KL}}(P, \bar{P}) - \log Z(q) \right) \tag{10}$$

Since $Z(q)$ doesn't depend on $P$, the minimum of (10) is achieved by minimizing the first KL term. With Gibbs' inequality that KL-divergence is minimized at 0 if and only if the two distributions are identical. Therefore, we have the optimal solution:

$$P_q^*(o) = \bar{P}(o|q) = \frac{1}{Z(q)} \pi_{\text{old}}^+(o|q) \exp(r_l(o)/\lambda) \tag{11}$$

$$= \frac{\pi_{\text{old}}^+(o|q) \exp(r_l(o)/\lambda)}{\mathbb{E}_{o \sim \pi_{\text{old}}^+(\cdot|q)} \exp(r_l(o)/\lambda)}. \tag{12}$$

## A.2  DETAILED HYPERPARAMETER SETTING

In this section, we provide detailed hyperparameter settings used in our experiments.

For all the methods, we employ the AdamW optimizer with a weight decay of 0.01 and set constant learning rate to $2\text{e}^{-6}$ for 1.5B model and $1\text{e}^{-6}$ for 7B model, following Li et al. (2025a). We set the batch size to 128 for each step of RL, the mini-batch size to 32 for each iteration of model update, and sample 8 responses per question for training. The generation budget is limited to 8k tokens for both training and evaluation. The temperature is set to 0.6 for training.

For RLOO-LP, we use RLOO advantage estimator and clip ratio $\epsilon = 0.2$. We tune their weight parameter $\alpha \in \{0.05, 0.1, 0.2\}$. Following their paper, we normalize its loss by the length of the response.

Table 2: Various reward designs for efficient reasoning, where $r_c(o|q) = \mathbb{I}(o \text{ is correct}) \in \{1, 0\}$ denotes correctness reward.

| Method | $r(o, q)$ |
|---|---|
| RLOO-LP (Arora & Zanette) | $r_c(o, q) - \alpha * r_c(o, q) * \sigma\left(\frac{|o| - \text{mean}\{|o_i|, r_c(o_i, q)=1\}}{\text{std}\{|o_i|, r_c(o_i, q)=1\}}\right)$ |
| ALP (Xiang et al.) | $r_c(o, q) - \beta * |o| * \max(\text{mean}\{r_c(o_i, q)\}, K^{-1})$ |
| HAPO (Huang et al.) | $r_c(o, q) + w * \max\left(\cos(\min(\frac{\pi}{2}\frac{|o|}{h(q)}, \pi)), c\right) r_c(o, q) + w * \min\left(\cos(\min(\frac{\pi}{2}\frac{|o|}{h(q)}, \pi)), 0\right)(1 - r_c(o, q))$ |
| L1-MAX (Aggarwal & Welleck) | $r_c(o, q) * \text{clip}(\alpha(L_T - |o|) + \delta), 0, 1)$ |
| SB (Yi et al., 2025) | $\alpha * r_c(o, q) - \beta * \text{abs}\left(|o| - L_{SOL}(q)\right)$, where $L_{SOL}(q) = \min\{|o_i|, r_c(o_i, q) = 1\}$ if at least one response is correct, $\text{mean}\{|o_i|\}$ otherwise |
| LASER-D (Liu et al., 2025b) | $r_c(o, q) + r_c(o, q) * \alpha \mathbb{I}(|o| \leq L_A)$ |

Table 3: Illustration of negative learning signal of correct outputs of existing reward designs.

| Method | Hyperparamter | Length | Correctness | Reward | Advantage |
|---|---|---|---|---|---|
| RLOO-LP | $\alpha$=0.4 | [1500, 1200, 1900, 2200, 2800, 2000, 3600, 6400, 1300, 1200] | [1, 1, 1, 1, 1, 1, 1, 1, 0, 0] | [0.87, 0.89, 0.85, 0.83, 0.79, 0.84, 0.74, 0.63, 0., 0. ] | [ 0.25, 0.27, 0.23, 0.21, 0.16, 0.22, 0.11, -0.02, -0.72, -0.72] |
| ALP | $\beta$=0.0001 | [1500, 1200, 1900, 2200, 2800, 2000, 3600, 6400, 1300, 1200] | [1, 1, 1, 1, 1, 1, 1, 1, 0, 0] | [ 0.88, 0.9, 0.85, 0.82, 0.78, 0.84, 0.71, 0.49, -0.1 , -0.1 ] | [ 0.74, 0.8, 0.65, 0.58, 0.46, 0.63, 0.28, -0.32, -1.92, -1.9 ] |
| HAPO | $w$=1, $c$=-0.7, $h$=1200 | [1500, 1200, 1900, 2200, 2800, 2000, 3600, 6400, 1300, 1200] | [1, 1, 1, 1, 1, 1, 1, 1, 0, 0] | [ 0.62, 1., 0.3, 0.3, 0.3, 0.3, 0.3, 0.3, -0.13, -0. ] | [ 0.99, 2.29, -0.1, -0.1, -0.1, -0.1, -0.1, -0.1, -1.57, -1.12] |
| L1-MAX | $\alpha$=0.0003, $L_T$=4000, $\delta$=0.5 | [1500, 1200, 1900, 2200, 2800, 2000, 3600, 6400, 1300, 1200] | [1, 1, 1, 1, 1, 1, 1, 1, 0, 0] | [1., 1., 1., 1. , 0.86 , 1., 0.62 ,0., 0., 0. ] | [0.8, 0.8, 0.8, 0.8 , 0.48, 0.8, -0.06, -1.48, -1.48, -1.48 ] |
| SB | $\alpha$=2, $\beta$=0.001 | [1500, 1200, 1900, 2200, 2800, 2000, 3600, 6400, 1300, 1200] | [1, 1, 1, 1, 1, 1, 1, 1, 0, 0] | [ 1.7, 2., 1.3, 1., 0.4, , 1.2, -0.4, -3.2, -0.1, 0.] | [ 0.92, 1.14, 0.64, 0.43, 0.01, 0.57, -0.56, -2.53, -0.35, -0.28] |
| LASER-D | $\alpha$=0.5, $L_A$=4000 | [1500, 1200, 1900, 2200, 2800, 2000, 3600, 6400, 1300, 1200] | [1, 1, 1, 1, 1, 1, 1, 1, 0, 0] | [1.5, 1.5, 1.5, 1.5, 1.5, 1.5, 1.5, 1., 0., 0. ] | [ 0.59, 0.59, 0.59, 0.59, 0.59, 0.59, 0.59, -0.25, -1.94, -1.94] |

For ALP, we follow their paper to use GRPO advantage estimator and normalize its loss by the total number of tokens. The KL Coefficient is set to 0.001. We tune their penalty weight $\beta \in \{1e^{-9}, 1e^{-8}, 1e^{-7}\}$ and and clip ratio $\epsilon = 0.2$

For HAPO, we follow their paper to use GRPO advantage estimator, set the cutoff $c = -0.7$, KL Coefficient to 0, and clip ratio $\epsilon = 0.2$. The loss is normalized by the length of the response. We tune their weight parameter $w \in \{0.01, 0.1, 1\}$.

For the proposed DRPO, we follow Li et al. (2025a) to set constraint value $\delta = 1e^{-4}$, penalty constant $\beta_0 = 1e^3$, $\tau = 10$. we tune regularization parameter $\lambda \in \{0.5, 0.2, 0.1\}$.

## A.3 LIMITATION OF INCORPORATING LENGTH REWARD WITH GROUP ADVANTAGE

In this part, we summarize different reward designs of existing baselines, which incorporate a length reward to encourage efficient reasoning, in Table 2. To illustrate the inherent limitation of incorporating length reward with group advantage, in Table 3, we provide detailed examples of how these reward designs fail to work with group advantage, resulting in misleading learning signals. Specifically, we follow the hyperparameters used in their paper to calculate the advantage with RLOO advantage estimator for RLOO-LP method and GRPO advantage estimator (i.e., Eqn. (2)) for other methods. RLOO advantage estimator is calculated as $A(o_i|q) = r(o_i|q) - \text{mean}(r(o_1|q), \cdots, r(o_{i-1}|q), r(o_{i+1}|q), \cdots, r(o_G|q))$. As indicated by red values in Table 3, all reward designs produce varying amounts of misleading learning signals. We see that HAPO suffers most, yielding incorrect learning directions in 6 out of 10 cases. This could help explain why HAPO exhibits larger performance degradation than other baselines in our experiments.

## A.4 EXPERIMENTS ON NON-MATHEMATICAL REASONING TASK

To evaluate DRPO's generalization on non-mathematical reasoning tasks, we conducted additional experiments on the logic puzzle reasoning task. Following Xie et al. (2025); Su et al. (2025), the training dataset is limited to 3 to 7-person logic puzzles with K&K logic puzzle dataset (Xie et al., 2024) and the test dataset contains 2 to 8-person puzzles. We train 1.5B models for all methods for 400 steps and conduct evaluation every 80 steps. As shown in Figure 5, DRPO method still exhibits a substantially better trade-off than all other baselines, demonstrating the generalizability of the proposed method to other tasks. Notably, DRPO significantly reduces the generation length

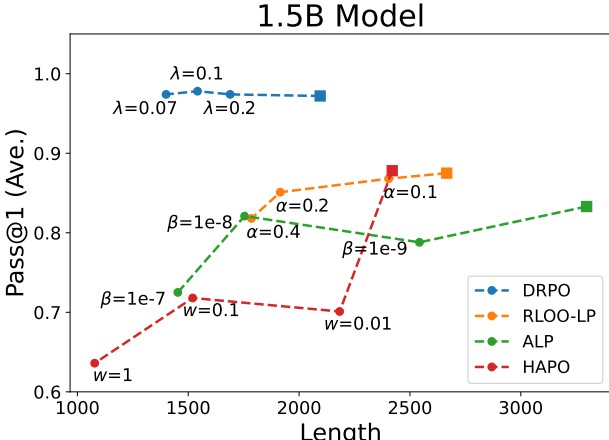

Figure 5: Comparison of performance-efficiency trade-off on logical puzzle reasoning task. Squares denote models trained with reference methods without length penalties, i.e., $\lambda$=+$\infty$ (corresponding to DisCO) for DRPO, $\alpha = 0$ for RLOO-LP, $\beta = 0$ (corresponding to GRPO) for ALP, $w = 0$ for HAPO. Triangles denote the models trained by other works.

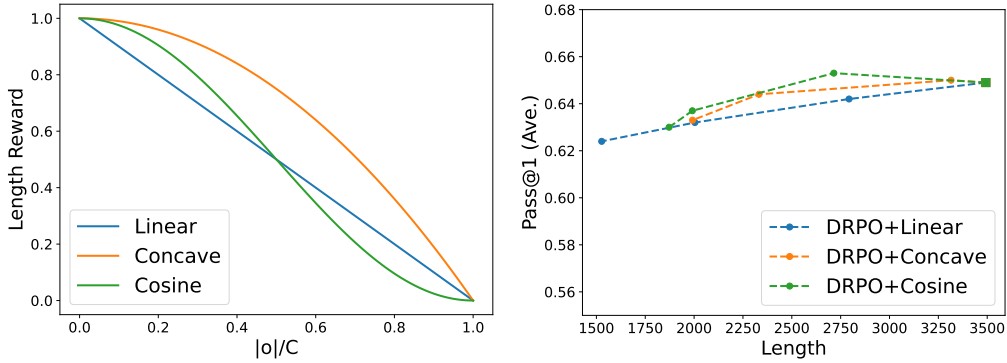

Figure 6: Ablation studies on different reward designs. Left is the length reward values with respect to $|o|/C$; Right is the performance-efficiency trade-offs for different reward designs on mathematical reasoning task. Square denotes the model trained without length penalties, i.e., $\lambda$=+$\infty$ (DisCO).

from 2095 to 1400 (33.2% length reduction) without any performance drop (from an accuracy of 0.972 to 0.974 ).

### A.5  ABLATION STUDY ON LENGTH REWARD DESIGN

In the main experiments, we adopted a simple length reward $r_l(o) = 1 - \frac{|o|}{C}$, where $C$ is a constant denoting maximum response length. To study the effect of different length reward designs in DRPO, we conduct experiments on mathmatical reasoning task with (1) a concave length reward (i.e., $r_l(o) = 1 - (\frac{|o|}{C})^2$) and (2) cosine length reward (i.e., $0.5 + 0.5\cos(\pi * \frac{|o|}{C})$), using a 1.5B model. For all reward design choices, we tune $\lambda$ in $\{0.5, 0.2, 0.1\}$. The results are summarized in Figure 6. We observe that all reward designs achieve competitive performance. The linear reward provides the broadest trade-off spectrum, while the concave and cosine rewards tend to yield higher accuracy at the cost of longer reasoning lengths. These findings suggest that developing more sophisticated length rewards is a promising direction for further improving DRPO.

Table 4: Detailed AES performance for 1.5B models.

| Method | GSM8K | MATH500 | OlympiadBench | AIME | Ave. |
|---|---|---|---|---|---|
| RLOO-LP ($\alpha$=0.1) | 0.078 | 0.021 | -0.423 | -0.738 | -0.172 |
| RLOO-LP ($\alpha$=0.2) | 0.246 | 0.325 | -0.279 | -0.694 | **-0.129** |
| RLOO-LP ($\alpha$=0.4) | 0.026 | -0.13 | -0.454 | -1.18 | -0.412 |
| ALP ($\beta$=1e-9) | 0.06 | 0.016 | -0.603 | -1.571 | **-0.387** |
| ALP ($\beta$=1e-8) | -0.189 | -0.088 | -0.91 | -2.229 | -0.602 |
| ALP ($\beta$=1e-7) | -1.019 | -0.518 | -1.887 | -3.774 | -1.4 |
| HAPO ($w$=0.01) | -0.209 | -0.348 | -0.879 | 0.202 | **-0.519** |
| HAPO ($w$=0.1) | -0.74 | -0.811 | -1.066 | -0.457 | -0.969 |
| HAPO ($w$=1) | -1.05 | -1.127 | -0.841 | -0.182 | -1.063 |
| DRPO ($\lambda$=0.5) | 0.296 | 0.21 | 0.152 | -0.143 | 0.093 |
| DRPO ($\lambda$=0.2) | 0.614 | 0.296 | 0.184 | -0.597 | 0.164 |
| DRPO ($\lambda$=0.1) | 0.662 | 0.332 | 0.098 | -0.729 | **0.178** |

Table 5: Detailed AES performance for 7B models.

| Method | GSM8K | MATH500 | OlympiadBench | AIME | Ave. |
|---|---|---|---|---|---|
| RLOO-LP ($\alpha$=0.1) | 0.391 | 0.23 | -0.051 | -0.532 | **-0.033** |
| RLOO-LP ($\alpha$=0.2) | 0.211 | 0.284 | -0.046 | -0.829 | -0.122 |
| RLOO-LP ($\alpha$=0.4) | 0.045 | -0.04 | -0.103 | -0.709 | -0.331 |
| ALP ($\beta$=1e-9) | 0.451 | 0.265 | -0.297 | -0.984 | **-0.134** |
| ALP $\beta$=1e-8) | 0.161 | -0.351 | -1.048 | -1.798 | -0.68 |
| ALP ($\beta$=1e-7) | -0.923 | -1.211 | -1.745 | -2.679 | -1.573 |
| HAPO ($w$=0.01) | -0.105 | -1.001 | -1.318 | -1.676 | **-1.08** |
| HAPO ($w$=0.1) | -0.999 | -1.483 | -1.252 | -1.197 | -1.42 |
| HAPO ($w$=1) | -1.483 | -1.407 | -1.344 | -1.503 | -1.6 |
| DRPO ($\lambda$=0.5) | 0.053 | 0.106 | -0.126 | -0.155 | -0.007 |
| DRPO ($\lambda$=0.2) | 0.439 | 0.303 | 0.015 | -0.33 | 0.115 |
| DRPO ($\lambda$=0.1) | 0.672 | 0.514 | 0.254 | -0.455 | **0.249** |

## A.6 DETAILED AES PERFORMANCE

In this part, we present detailed AES performance for each method in Table 4, 5, and 6, where bold values denote the best AES performance for each method. We observe that all baseline methods yield negative AES scores for almost all settings, underscoring their inefficiency in preserving performance while reducing reasoning length. In contrast, DRPO consistently achieves positive AES scores for most cases, demonstrating its effectiveness in improving reasoning efficiency without sacrificing performance.

## A.7 CASE STUDY

We analyze the reasoning path of DRPO, compared with DisCO, which corresponds to DRPO with $\lambda = +\infty$. Figure 7 shows the reasoning paths on an easy prompt, where DRPO reaches the correct answer with clear reasoning in only 89 tokens, achieving a 6× reduction compared to the 526 tokens required by DisCO. Although DisCO also produces the correct answer, its reasoning is highly redundant, including four occurrences of reflection words (e.g., wait) and three repeated answers. By contrast, DRPO's reasoning involves only a single reflection and one correct answer.

We further evaluate the reasoning efficiency on a more challenging prompt in Figure 8 and Figure 9. As shown in Figure 8, DisCO requires 4497 tokens, about 10 times longer than the 455 tokens of DRPO, to solve the question. Moreover, the reasoning of DisCO contains 78 reflection words and 19 answers, six of which are incorrect. In comparison, DRPO produces only two correct answers and

Table 6: Detailed AES performance for 8B models.

| Method | GSM8K | MATH500 | OlympiadBench | AIME | Ave. |
|---|---|---|---|---|---|
| RLOO-LP ($\alpha$=0.1) | 0.378 | 0.198 | 0.282 | 0.295 | 0.212 |
| RLOO-LP ($\alpha$=0.2) | 0.515 | 0.203 | 0.313 | 0.315 | **0.251** |
| RLOO-LP ($\alpha$=0.4) | 0.064 | -0.126 | 0.127 | 0.392 | -0.119 |
| ALP ($\beta$=1e-9) | 0.476 | 0.215 | 0.053 | -0.67 | **-0.01** |
| ALP $\beta$=1e-8) | 0.22 | -0.252 | -0.686 | -2.405 | -0.584 |
| ALP ($\beta$=1e-7) | -2.139 | -1.773 | -1.81 | -3.733 | -2.289 |
| HAPO ($w$=0.01) | -0.021 | -0.17 | -0.251 | -0.755 | -0.413 |
| HAPO ($w$=0.1) | 0.197 | -0.264 | -0.445 | -0.673 | **-0.366** |
| HAPO ($w$=1) | -0.23 | -0.667 | -0.542 | -1.579 | -0.775 |
| DRPO ($\lambda$=0.5) | 0.29 | 0.172 | 0.098 | 0.091 | 0.122 |
| DRPO ($\lambda$=0.2) | 0.607 | 0.435 | 0.313 | 0.068 | 0.296 |
| DRPO ($\lambda$=0.1) | 0.624 | 0.449 | 0.292 | 0.04 | **0.297** |

---

**Algorithm 1** Decoupled Reward Policy Optimization (DRPO)

---

1: **Input:** Initial policy model $\pi_0$, reward function $r$, question set $\mathcal{D}$, hyperparameter $\delta, \beta, \tau, \lambda$.
2: Policy model $\pi_\theta = \pi_0$
3: **for** Step $= 1, \cdots, T$ **do**
4:     Sample a batch of questions $\mathcal{B}$ from $\mathcal{D}$
5:     Update the old policy model $\pi_{\text{old}} = \pi_\theta$
6:     For each question $q \in \mathcal{B}$, sample $n$ responses $\{o_i\}_{i=1}^n \sim \pi_{\text{old}}(\cdot|q)$ denoted by $S_q$ and partition it into $S_q^+$ and $S_q^-$ based on correctness rewards $r(o_i|q) \in \{0, 1\}$
7:     **for** minibatch $\mathcal{B}_m \in \mathcal{B}$ **do**
8:         Compute KL divergence estimator by

$$\hat{\mathbb{D}}_{KL}(\theta) = \frac{1}{\sum_{q \in \mathcal{B}_m} \sum_{o \in S_q} |o|} \sum_{q \in \mathcal{B}_m} \sum_{o \in S_q} \sum_{t=1}^{|o|} \log \frac{\pi_{\theta_{old}}(o_t|q, o_{<t})}{\pi_\theta(o_t|q, o_{<t})}$$

9:         Compute gradient estimator of the objective in Eqn. 7 by

$$G_1 = \frac{1}{|\mathcal{B}_m|} \sum_{q \in \mathcal{B}_m} \left( \sum_{o \in S_q^+} \frac{\exp(r_l(o)/\lambda)}{\sum_{o^* \in S_q^+} \exp(r_l(o^*)/\lambda)} \nabla s_\theta(o, q) - \nabla \left( \tau \log \sum_{o' \in S_q^-} \exp\left(\frac{s_\theta(o', q)}{\tau}\right) \right) \right)$$

10:        Compute gradient estimator of a penalty function of the constraint by $G_2 = 2\beta_0[\hat{\mathbb{D}}_{KL}(\theta) - \delta]_+\nabla\hat{\mathbb{D}}_{KL}(\theta)$
11:        Update $\pi_\theta$ with Adam-W using the gradient estimator $G = G_1 + G_2$
12:     **end for**
13: **end for**

---

five reflection words as seen in Figure 9. It is worth noticing that reasoning from DisCO explicitly says that 'but I'm not 100% sure' before concluding the final answer. It suggests that excessive back-and-forth in the reasoning, indicated by frequent use of reflection words, may result in less confident answers.

LLM USAGE

We use LLM only to help correct grammar errors and polish writing.

**Prompt 1**

A robe takes 2 bolts of blue fiber and half that much white fiber. How many bolts in total does it take? Let's think step by step and output the final answer within \boxed{}.

**Answer from DisCO (DRPO $\lambda = +\infty$)** 526 tokens

Okay, so I have this problem here: A robe takes 2 bolts of blue fiber and half that much white fiber. I need to figure out how many bolts in total it takes. Hmm, let me break this down step by step.

First, the problem says the robe uses 2 bolts of blue fiber. That's straightforward. Now, it also says it takes half that much white fiber. So, I need to find out how much white fiber is that.

Let me write this out. Blue fiber is 2 bolts. White fiber is half of that. So, I can calculate the white fiber as:

White fiber = (1/2) * Blue fiber
White fiber = (1/2) * 2

Hmm, let me do that math. Half of 2 is 1, so white fiber is 1 bolt.

Now, the total number of bolts is the sum of blue fiber and white fiber. So, that would be:

Total bolts = Blue fiber + White fiber
Total bolts = 2 + 1

Which equals 3. So, the total number of bolts needed is 3.

Wait, let me just make sure I didn't skip any steps or make any mistakes. So, blue fiber is 2, white fiber is half of that, so 2 divided by 2 is indeed 1. Adding them together, 2 + 1 is 3. Yeah, that seems right.

I don't think I need to consider any other factors here, like mixing them or something. The problem just says it takes 2 bolts of blue and half that much white. So, 2 blue and 1 white, totaling 3.

Wait, another way to think about it: If half that much white is 1 bolt, then 2 blue plus 1 white is 3 total. Yeah, that's correct.

I don't think I need to overcomplicate this. It's pretty straightforward. So, the total number of bolts is 3.

**Final Answer**
The total number of bolts needed is \boxed{3}.
</think>

The robe takes 2 bolts of blue fiber. The white fiber required is half of that, which is:

\[
\text{White fiber} = \frac{1}{2} \times 2 = 1 \text{ bolt}
\]

Adding the blue and white fibers together:

\[
\text{Total bolts} = 2 + 1 = 3
\]

The total number of bolts needed is \boxed{3}.

**Answer from DRPO ($\lambda = 0.1$)** 89 tokens

Okay, so I need to figure out how many total bolts a robe takes. It says the robe takes 2 bolts of blue fiber and half that much white fiber. Wait, so blue is 2 bolts, white is half of blue, which is 1 bolt. So total is 2 +1=3 bolts. That seems right.
</think>

The total number of bolts needed is 3.

$\boxed{3}$

Figure 7: Example reasoning for Prompt 1 from DisCo (DRPO $\lambda = +\infty$) and DRPO ($\lambda = 0.1$). Words with green color are answers, and with blue color are reflection words. DRPO reaches the correct answer with clear reasoning in only 89 tokens, achieving a 6× reduction compared to the 526 tokens required by DisCO.

**Prompt 2**

Josh decides to try flipping a house. He buys a house for $80,000 and then puts in $50,000 in repairs. This increased the value of the house by 150%. How much profit did he make? Let's think step by step and output the final answer within \boxed{}.

**Answer from DisCO (DRPO with $\lambda = +\infty$)** 4497 tokens

Okay, so Josh bought a house for $80,000. Then he spent $50,000 on repairs. After that, the value of the house increased by 150%. I need to figure out how much profit he made. Let me break this down step by step. First, the original price of the house was $80,000. He spent $50,000 on repairs, so his total investment becomes: $80,000 (purchase) + $50,000 (repairs) = $130,000 Wait, but the problem says that the increase in value is 150%. So, I need to calculate the new value of the house after the increase. Hmm, the increase is 150% of the original purchase price? Or is it 150% of the total investment after repairs? Let me read the problem again: "This increased the value of the house by 150%." So, the increase is 150% of the original house price. Wait, no. The original value is $80,000, but he's already invested $130,000. Wait, maybe I should clarify. The problem says he bought the house for $80,000, then spent $50,000 on repairs. Then, the value of the house increased by 150%. I think that the increase is 150% on top of the total investment. So, the new value is the total investment plus 150% of that. Wait, let's represent this mathematically. Let me denote: Original cost: $80,000 Repairs: $50,000 Total cost: $80,000 + $50,000 = $130,000 The value increased by 150%. So, the increase is 150% of $130,000? Or is it 150% of the original house price? The problem says "increased the value of the house by 150%." So, the increase is 150% of the original value before repairs? Wait, no. It's probably the increase is 150% of the total investment. Wait, let's read the problem again: "Josh buys a house for $80,000 and then puts in $50,000 in repairs. This increased the value of the house by 150%. How much profit did he make?" So, he spent $80k and then $50k, so total $130k. Then the value increased by 150%, meaning the new value is 130k plus 150% of 130k? Wait, that can't be. Wait, maybe the increase is 150% on the repairs or something else. Wait, perhaps it's better to think that the increase in value is 150% of the original purchase price. Wait, no. Wait, the problem says "this increased the value of the house by 150%." Wait, maybe the total value after repairs is the original price plus 150% of the original price. Wait, the problem says "this increased the value by 150%." So, if the original value was $80k, then the new value is $80k + 150% of $80k? Wait, that would be: $80k + 1.5 * $80k = $80k + $120k = $200k. But that seems high. Wait, perhaps the increase is 150% of the total investment. So, total investment was $130k, and it increased by 150%, so the increase is 1.5*130k = $195k. So, the new value is $130k + $195k = $325k. Then, the total profit would be $325k - $80k (original) - $50k (repairs) = $325k - $130k = $195k. But that seems too high. Wait, maybe I need to clarify: Original price: $80,000 Repairs: $50,000, so total investment: $130,000 After repairs, the value increased by 150%. So, the increase is 150% of the total investment. So, the increase is 1.5 * 130,000 = 195,000. So, the new value is 130,000 + 195,000 = 325,000. Then, his total outflow was 80,000 + 50,000 = 130,000. His total inflow was 325,000. So, his profit is 325,000 – 130,000 = 195,000. Wait, but that seems like a lot. Alternatively, maybe the increase is 150% on top of the original purchase price. So, original purchase was $80,000, increased by 150% would be 80,000 * 1.5 = 120,000. So, the new value is 80,000 + 120,000 = 200,000. But he already put in 50k, so his total outflow is 130k. So, profit is 200k - 130k = 70k. Wait, this is conflicting. Wait, let's read the problem statement again: Josh buys a house for $80,000 and then puts in $50,000 in repairs. This increased the value of the house by 150%. How much profit did he make?" So, the process: 1. Buy house: $80,000 2. Repair: $50,000 3. Value increased by 150%: So, the increase is 150% of the total value after repair. Wait, let's formalize: Let V be the value after repairs. He bought it for 80k, spent 50k on repairs, so total outlay: 130k. The value V increased by 150%, so: V + 150% of V = total outlay? Wait, I think I need to model it differently. The value of the house increased by 150%, meaning that the new value is 150% more than the original purchase price? Wait, the problem says "this increased the value of the house by 150%." So, if the original value before repairs was 80k, then after repairs, it's 80k + 150% of 80k? Wait, that would be: Value after repairs = 80k + (1.5 * 80k) = 80k + 120k = 200k But he had already spent 50k on repairs, so his total outlay is 80k + 50k = 130k. So, his total gain is 200k - 130k = 70k profit. So, but the problem says "this increased the value of the house by 150%." So, the increase is 150% of the original value? Or 150% of the current value? Wait, perhaps the increase is 150% on top of the purchase price, so: Original purchase: 80k Repairs: 50k So, total outlay: 130k Value after repairs: original + (150% of 130k) = 130k + 195k = 325k Wait, but that seems too high. Alternatively, maybe the increase is 150% of the original value, so 80k * 1.5 = 120k. So, the new value is 130k (after repair) + 120k = 250k? Wait, I'm getting confused. Wait, let's try another approach. Let me denote V as the new value after repairs. The increase is 150% of something. The problem says "this increased the value of the house by 150%." So, the increase is 150% of the original house price, which is 80k. So, the increase is 1.5*80k=120k. So, the new value is 80k + 120k=200k. But Josh already put in 50k on repairs, so his total outlay is 80k +50k=130k. So, his profit is 200k -130k=70k. Wait, but that's the same as before. Wait, another interpretation: The increase is 150% on top of his investment. His investment was 130k, so the increase is 1.5*130k=195k. So, new value is 130k +195k=325k. Profit is 325k -130k=195k. Wait, but the problem says "this increased the value of the house by 150%." So, the value went up by 150% of its original value? Or of its current value? Wait, the problem is ambiguous in that sense. Let's see. It's possible that the total increase is 150% of the original purchase price. So, original price is 80k, increase is 150% of 80k=120k, so new value is 80k +120k=200k. Then, his outlay was 80k+50k=130k, so profit is 200k -130k=70k. Alternatively, the increase is 150% of the current value. Current value after repairs is 80k+50k=130k. So, increase is 150% of 130k=195k, new value would be 130k +195k=325k. Profit would be 325k -130k=195k. Wait, but the problem says "this increased the value of the house by 150%." So, the value increased by 150%, meaning that the new value is 150% more than the original. Wait, the original value is 80k, so 80k +150% of 80k=80k +120k=200k. Wait, that's another way. So, the increase is 150% of the original value, not the current value. So, in this case, the new value is 80k +150% of 80k=200k. His total outlay was 80k+50k=130k. So, profit is 200k -130k=70k. Wait, but the problem says "increased the value of the house by 150%." So, perhaps the value went up by 150%, meaning the new value is 100% +150%=250% of the original. Wait, that can't be because the original was 80k, and he spent 50k on repairs. Wait, I'm getting tangled here. Wait, maybe let's model it step by step: 1. Josh buys the house for 80k. 2. He then spends $50k on repairs, so total expenditure is $130k. 3. The value of the house increases by 150%, so the increase is 150% of the original purchase price? Or the total expenditure? Wait, the problem says "increased the value of the house by 150%." So, the increase is 150% of the current value? Or the original value. Wait, I think the problem is that the increase is 150% of the original purchase price, so the increase is 150% of $80k, which is $120k. So, the new value is 80k +120k=200k. His total expenditure was 80k +50k=130k. So, his profit is 200k -130k=70k. Wait, but the problem says "increased the value of the house by 150%." So, if the original value was 80k, then 150% of that is 120k, so the new value is 80k +120k=200k. So, profit is 200k - 130k=70k. Wait, but another way: Maybe the increase is 150% of the total expenditure. So, 150% of $130k is $195k, so new value is 130k +195k=325k. Profit is 325k -130k=325k. Wait, but that seems high. I think the problem is that the increase is 150% of the original purchase price, so 150% of 80k=120k, making the new value 80k +120k=200k. Hence, profit is 200k -130k=70k. Wait, but the problem says "increased the value of the house by 150%." So, the new value is 150% of the original. Wait, no. Wait, maybe the value went up by 150%, meaning the new value is 100% +150%=250% of the original. So, 250% of 80k=200k, same as before. Hence, profit is 200k -130k=70k. Wait, but 150% increase on the value after repairs. Wait, this is getting too tangled. Perhaps, the correct approach is: Let V be the value after repairs. The increase is 150%, so V + 1.5V = total outlay? Wait, no. Wait, the total outlay was 80k +50k=130k. The value after repairs increased by 150%, so V - 150% of V= original? Wait, I'm not getting this. Wait, perhaps the problem is that the value increased by 150% of the original purchase price. So, the increase is 150% of 80k=120k. So, the new value is 80k +120k=200k. Hence, profit is 200k -130k=70k, that seems reasonable. Let me check the steps: 1. Buy for 80k. 2. Repair: 50k, total spent 130k. 3. Value increased by 150% of original: 150% of 80k=120k, so new value 80k +120k=200k. 4. So, profit is 200k -130k=70k. Hence, Josh made a profit of $70,000. Wait, but I should make sure that the increase is 150% of the original or the current value. Wait, the problem says "this increased the value of the house by 150%." So, the increase is 150% of the original value, so 150% of 80k=120k. So, new value is 80k +120k=200k. Hence, profit is 200k -130k=70k. Hence, the answer is $70,000. Wait, but let me check this once more. Original purchase: 80k Repairs: 50k, total:130k Value increased by 150%: So, the value becomes 130k +150% of 130k=130k +195k=325k Wait, that's another way. Wait, so the increase is 150% of the current value, which is 150% of 130k=195k, so new value is 130k +195k=325k. Hence, profit is 325k -130k=195k. Wait, but which interpretation is correct? The problem says "increased the value of the house by 150%." So, it's ambiguous whether it's 150% of the original or 150% of the current. Wait, if it's 150% of the original, it's 120k increase, new value 200k. profit 70k. If it's 150% of the current, it's 195k increase, new value 325k, profit 195k. Given that the problem says "this increased the value of the house by 150%," I think it's 150% of the original purchase price. So, 150% of 80k=120k, new value 200k. Hence, profit is 200k -130k=70k. Wait, but wait, another thought: The "increase" is 150% of the total investment. Wait, total investment was 130k, 150% of that is 195k, so new value is 130k +195k=325k, profit 325k -130k=195k. I think I need to settle on one answer. Wait, maybe the problem is that the value increased by 150% of the original purchase price, which is 80k*1.5=120k, so new value is 80k+120k=200k. Hence, profit is 200k - (80k+50k)=200k -130k=70k. Hence, the answer is $70,000. Wait, but I'm not 100% sure, but given the problem statement, I think that's the answer. **Final Answer** \boxed{70000}

</think> Josh bought a house for $80,000 and spent $50,000 on repairs, totaling $130,000. The value of the house increased by 150%, meaning the new value is 80,000 + 150% of 80,000 = 200,000. The profit is calculated as the new value minus the total investment: 200,000 - 130,000 = 70,000. Thus, Josh made a profit of \boxed{70000}.

Figure 8: Example reasoning for Prompt 2 from DisCo (DRPO $\lambda = +\infty$). Words in green/red color are correct and incorrect answers, respectively. Blue words are reflection words. The reasoning of DisCO contains 78 reflection words and 19 answers, six of which are incorrect.

**Prompt 2**

Josh decides to try flipping a house. He buys a house for $80,000 and then puts in $50,000 in repairs. This increased the value of the house by 150%. How much profit did he make? Let's think step by step and output the final answer within \\boxed{}.

**Answer to prompt 2 from DRPO ($\lambda = 0.1$)**   455 tokens

Okay, so Josh buys a house for $80,000 and spends another $50,000 on repairs. So total cost is 80k +50k=130k.

Then the house's value increased by 150%. Wait, increased by 150% means it's now 100%+150%=250% of the original value.

Wait, original cost was $80,000, but after repairs, the total investment is $130,000, right? Wait, maybe I should clarify: when they say the house's value increased by 150%, is it based on the original purchase price or the total cost?

Hmm, usually, such problems consider the increase based on the cost basis. So if the house increased by 150%, that's 150% of the original purchase price, which was $80,000.

So 150% of 80k is 1.5*80k=120k. So the value went up by 120k, so new value is 80k+120k=200k.

Wait, but he spent 50k on repairs. So his total investment was 130k, and he sold it for 200k. So profit is 200k - 130k=70k.

Wait, let me double-check:

Purchase price:80k

Repairs:50k

Total cost:130k

Value increased by150%: so 150% of 80k=120k, so new value=80k+120k=200k.

He sold it for 200k, so profit=200k-130k=70k.

Yes, so profit is $70,000.
</think>

Josh made a profit of $70,000.

The answer is \boxed{70000}.

Figure 9: Example reasoning for Prompt 2 from DRPO ($\lambda = 0.1$). Words in green/red color are correct and incorrect answers, respectively. Blue words are reflection words. DisCO uses 4497 tokens, about 10 times longer than the 455 tokens of DRPO.

