# OpenReview forum: "DRPO: Efficient Reasoning via Decoupled Reward Policy Optimization"
_ICLR.cc/2026/Conference — ICLR 2026 Poster_

### Official Review · Reviewer_Wot1 · 2025-10-25

**Soundness:** 2
**Presentation:** 3
**Contribution:** 3
**Rating:** 4
**Confidence:** 4

**Summary:**

This paper proposes DRPO.
DRPO tackles overthinking in large reasoning models by decoupling length-based rewards for correct answers from incorrect ones, preventing GRPO’s group-relative advantage from misclassifying verbose correct solutions as negative.
It introduces a KL-regularized, closed-form reweighting of on-policy correct samples that favors concise reasoning without ever flipping their learning signal to negative.
On four mathematical benchmarks, DRPO trims output length up to 77 % while losing only ~1 % accuracy, outperforming six strong baselines that sacrifice 4–7 % for smaller reductions.

**Strengths:**

* The paper identifies a subtle but critical flaw in GRPO’s group-relative advantage when length penalties are added, and proposes the first principled solution that decouples the normalization of correct and incorrect samples, a conceptual advance that is both novel and generalizable to other composite rewards.
* Extensive comparisons on four standard math-reasoning datasets show DRPO consistently dominates six recent baselines across 1.5 B and 7 B scales; ablations, AES metrics, and per-difficulty breakdowns confirm that the gains hold under rigorous evaluation, while a closed-form derivation guarantees computational efficiency without extra data.
* Good writing and easy to follow

**Weaknesses:**

* All experiments stop at 7 B parameters and four math domains; without data on >=30 B models or non-mathematical reasoning (code, science, commonsense).
* The length reward $r_l(o)=1−\frac{|o|}{C}$ is essentially linear; ablations with concave, step-wise or difficulty-aware penalties (e.g. LASER-D) are missing.

**Questions:**

* How does the performance of DRPO compare to the original DisCo algorithm? Adding relevant comparisons in Figure 3 would facilitate better comparison.
* Is DRPO still effective for step-wise rewards (i.e., process-level reward)?
* Could DRPO generalize to different families of LLMs?
* As mentioned in weakness, how DRPO performs on non-mathematical reasoning tasks?

---

> ### Author Response · Authors · 2025-11-20
> **Part 1/2**
>
> We thank the reviewer for providing helpful comments on improving our paper. Below we would like to answer raised questions.
>
> **Q1:** All experiments stop at 7 B parameters and four math domains. Could DRPO generalize to >=30 B models or different families of LLMs? How does DRPO perform on non-mathematical reasoning tasks?
>
> **A:** We thank the reviewer for the constructive comment. Due to computational constraints, we are not able to run >=30 B models. However, to assess the generalization ability of DRPO across **different LLMs**, we further validated our method on a model from another LLM-family, i.e., DeepSeek-R1-Distill-Llama-8B, comparing with other baselines. Similar to previous settings, we train models for all methods for 1000 steps and conduct evaluation every 200 steps. The results are presented in Figure 3 (right), Table 1 and Figure 4(c) in the revised manuscript. We also summarized the main results in the following Table. We observed that our proposed DRPO method still outperforms other baselines by a large margin, demonstrating the generalizability of the proposed method to other models.
>
> |  | pass@1/length | pass@1/length | pass@1/length | pass@1/length |
> |-|-|-|-|-|
> |DRPO|0.703/3394 ($\lambda = +\infty$)|0.705/3010 ($\lambda = 0.5$)|0.697/2101 ($\lambda = 0.2$)|0.690/1758 ($\lambda = 0.1$)|
> |RLOO-LP|0.643/3363 ($\alpha=0$)|0.643/2651 ($\alpha=0.1$)|0.619/2249 ($\alpha=0.2$)|0.611/2091 ($\alpha=0.4$)|
> | ALP|0.661/3646 ($\beta=0$)|0.637/2358 ($\beta=1e^{-9}$)|0.584/1527 ($\beta=1e^{-8}$)|0.467/1291 ($\beta=1e^{-7}$)|
> | HAPO|0.647/3521 ($w=0$)|0.594/2091 ($w=0.01$)|0.595/1981 ($w=0.1$)|0.564/1734 ($w=1$)|
>
> In addition, to evaluate DRPO’s performance on **non-mathematical reasoning** tasks, we conducted further experiments on the logic puzzle reasoning task. Using K&K logic puzzle dataset [r1], we train 1.5B models for all methods for 400 steps and conduct evaluation every 80 steps. The results are summarized in the following Table and Figure 5 in the revised manuscript. We observed that DRPO method still exhibits a substantially better trade-off than all other baselines, demonstrating the generalizability of the proposed method to other tasks. Notably, DRPO effectively reduces the generation length from 2095 to 1400 (33.2\% length reduction) without any performance drop (from an accuracy of 0.972 to 0.974 ).
>
> | | pass@1/length | pass@1/length | pass@1/length | pass@1/length |
> |-|-|-|-|-|
> |DRPO|0.972/2095 ($\lambda = +\infty$)|0.974/1689 ($\lambda = 0.2$)|0.978/1542 ($\lambda = 0.1$)|0.974/1400 ($\lambda = 0.07$)|
> |RLOO-LP|0.875/2666 ($\alpha=0$)|0.868/2404 ($\alpha=0.1$)|0.851/1915 ($\alpha=0.2$)|0.818/1785 ($\alpha=0.4$)|
> |ALP|0.833/3297 ($\beta=0$)|0.788/2543 ($\beta=1e^{-9}$)|0.821/1754 ($\beta=1e^{-8}$)|0.725/1453 ($\beta=1e^{-7}$)|
> | HAPO|0.878/2420 ($w=0$)|0.701/2183 ($w=0.01$)|0.718/1520 ($w=0.1$)|0.636/1078 ($w=1$)|
>
>
> **Q2:** Ablations with other forms of length reward are missing.
>
> **A:** Our proposed framework is **agnostic to the choice of length reward design**. To examine how different reward shapes affect performance, we conducted additional experiments using a 1.5B model with the following alternatives:
> (1) a **concave** length reward $r_l(o) = 1 - \left(\frac{|o|}{C}\right)^2$; and
> (2) a **cosine** length reward $0.5 + 0.5 \cos\left(\pi \frac{|o|}{C}\right)$.
> For each reward function, we tuned $\lambda \in \{ 0.5, 0.2, 0.1 \} $. The results are shown in Figure 6 of the revised manuscript and summarized in the table below. We observe that all reward designs achieve competitive performance. The **linear** reward provides the broadest trade-off spectrum, while the **concave** and **cosine** rewards tend to yield higher accuracy at the cost of longer reasoning lengths. These findings suggest that developing more sophisticated length rewards is a promising direction for further improving DRPO.
>
> |  | pass@1/length  | pass@1/length  | pass@1/length  | pass@1/length  |
> | - | - | - | - | -|
> | | $\lambda=+\infty$ | $\lambda=0.5$ | $\lambda=0.2$ | $\lambda=0.1$ |
> | Linear  | 0.649/3493     | 0.642/2793     | 0.632/2004     | 0.624/1527     |
> | Concave | 0.649/3493     | 0.650/3315     | 0.644/2332     | 0.633/1992     |
> | Cosine  | 0.649/3493     | 0.653/2715     | 0.637/1991     | 0.630/1872     |
>
> **Q3:** How does the performance of DRPO compare to the original DisCo algorithm? Adding relevant comparisons in Figure 3 would facilitate better comparison.
>
> **A:** In Figure 3 and 4, $λ=+\infty$ indeed corresponds to the reference method DisCO, which does not employ length reward during training. We made it clear in the revised manuscript.
>
> [r1] Xie, Chulin, et al. "On memorization of large language models in logical reasoning." 2024.

---

> > ### Author Response · Authors · 2025-11-20
> > **Part 2/2**
> >
> > **Q4:** Is DRPO still effective for step-wise rewards (i.e., process-level reward)?
> >
> > **A:** Yes, it is possible!  DRPO can apply to other rewards like process-level reward. For example, assum each response $o$ contains $m$ steps $\{s^1,...,s^m\}$ with step-wise rewards $r_s(s^1),...,r_s(s^m)$, as well as accuracy reward $r(o)\in\{0, 1\}$. One simple solution is directly replacing length reward $r_l(o)$ with process-level reward $r_s(o)=\frac{1}{m}\sum_m r_s(s^m)$ in Eqn.(5), which maximizes process-level reward with a KL regularization. This results in a final objective similar to Eqn.(7), encouraging the model to produce positive responses with high process-level reward.

---

### Official Review · Reviewer_Z7wp · 2025-10-31

**Soundness:** 2
**Presentation:** 3
**Contribution:** 2
**Rating:** 4
**Confidence:** 5

**Summary:**

This paper proposes a novel RL objective to enhance reasoning efficiency in LLMs by encouraging shorter reasoning paths. The authors observe that applying a length penalty reward may cause long but correct responses to receive negative advantages, inadvertently promoting unnecessarily long reasoning trajectories. To address this issue, they propose a decoupled advantage computation that separately handles positive and negative responses, ensuring positive advantages for high-quality responses. Empirical studies on DeepSeek-R1-Distill-Qwen-1.5B and 7B models across multiple mathematical reasoning benchmarks verify the effectiveness of the proposed approach in reducing reasoning length while maintaining accuracy.

**Strengths:**

1. The paper is well written and clearly organized.

2. The analysis of positive responses receiving negative advantages is insightful and leads naturally to the proposed DRPO formulation.

3. The proposed DRPO method outperforms other efficient reasoning approaches by a substantial margin.

**Weaknesses:**

1. The authors claim that redundant reasoning arises from positive responses with negative advantages. However, even the vanilla GRPO without a length penalty still exhibits redundant reasoning, despite all positive responses receiving positive advantages. More in-depth analysis is needed to substantiate the claim that “positive responses with negative advantages greatly harm reasoning efficiency.”


2. The experiments lack a comparison against the vanilla GRPO baseline. In the abstract, the authors mention a 1.1% performance drop of DRPO compared to the baseline. If DRPO exhibits a larger accuracy decline than GRPO, it would weaken the claimed significance of the proposed efficiency improvement.


3. In my opinion, measuring reasoning efficiency solely by token count is not comprehensive. An efficient response should be logical, step-by-step, and free from redundant reasoning, rather than simply shorter in length.

**Questions:**

Please see Weaknesses.

---

> ### Author Response · Authors · 2025-11-20
>
> We thank the reviewer for dedicating the time to providing helpful feedback on our paper. Below we would like to answer the raised questions.
>
> **Q1:** More in-depth analysis is needed to substantiate the claim that “positive responses with negative advantages greatly harm **reasoning efficiency.**”
>
> **A:** We believe this is a **misunderstanding**. This is not what we claimed. Our argument is that positive responses with negative advantages discourage valid reasoning, which in turn harms **reasoning accuracy**. In our work, “efficiency” specifically refers to the length of reasoning tokens, not accuracy. This issue of positive responses receiving negative advantages in baseline methods explains the dramatic accuracy degradation observed in Figure 3 for GRPO-based length-penalty approaches such as ALP and HAPO.
>
>
> **Q2:** The experiments lack a comparison against the vanilla GRPO baseline. In the abstract, the authors mention a 1.1% performance drop of DRPO compared to the baseline. If DRPO exhibits a larger accuracy decline than GRPO, it would weaken the claimed significance of the proposed efficiency improvement.
>
> **A:** We did! In fact, ALP with $\beta = 0$ (the square marker in Figures 3 and 4) corresponds to the **naive GRPO method**, since ALP is built on GRPO and reduces to GRPO when the length-penalty weight $\beta$ is set to zero. We have clarified this in the revision (see the caption of Figure 3 and 4). The “baseline” mentioned in the abstract refers to **DISCO**. As summarized in the table below, **DRPO consistently outperforms GRPO with different model sizes**.
>
> |            |                         | GRPO  | DRPO($\lambda=0.5$)| DRPO($\lambda=0.2$) | DRPO($\lambda=0.1$) |
> | ---------- | ----------------------- | ----- | ----- | ----- | ----- |
> | 1.5B model | Pass@1 | 0.606 | 0.642 | 0.632 | 0.624 |
> |            | Length                  | 3494  | 2793  | 2004  | 1527  |
> | 7B model   | Pass@1 | 0.710 | 0.727 | 0.719 | 0.714 |
> |            | Length                  | 3112  | 2823  | 2118  | 1502  |
> | 8B model   | Pass@1 | 0.661 | 0.705 | 0.697 | 0.690 |
> |            | Length                  | 3646  | 3010  | 2101  | 1758  |
>
>
>
> **Q3:** In my opinion, measuring reasoning efficiency solely by token count is not comprehensive. An efficient response should be logical, step-by-step, and free from redundant reasoning, rather than simply shorter in length.
>
> A: We agree. Efficient reasoning remains an open research problem and requires deeper investigation. Nevertheless, token count is a meaningful and direct metric because it directly reflects reasoning computation cost and response latency. We believe it will serve as a cornerstone for developing more advanced efficient reasoning methods. Importantly, DRPO can be applied to any efficiency metric once it is well-defined, simply by replacing $r_l(o)$ in Eqn. (5). Exploring alternative efficiency measures is an exciting direction that we leave for future work.

---

### Official Review · Reviewer_ziRG · 2025-10-31

**Soundness:** 4
**Presentation:** 3
**Contribution:** 3
**Rating:** 8
**Confidence:** 4

**Summary:**

The paper proposes DRPO, a reinforcement learning framework to improve reasoning efficiency. It identifies a flaw in GRPO where length penalties distort advantage estimation and introduces a decoupled reward formulation that separates positive and negative samples. Experiments on multiple reasoning benchmarks show that DRPO significantly shortens reasoning chains with minimal accuracy loss.

**Strengths:**

1. Identifies and formalizes a core issue in GRPO under multi-objective rewards.
2. Proposes a principled decoupling mechanism with a closed-form optimization solution.
3. Demonstrates substantial efficiency gains with competitive accuracy across multiple benchmarks.

**Weaknesses:**

1. Evaluation focuses only on math reasoning; unclear applicability to other domains (e.g., coding, logic).
2. The evaluation relies mainly on **Pass@1**, which can be unstable for reasoning tasks; additional metrics (e.g., Pass@k) would strengthen the empirical validation.

**Questions:**

1. Could the decoupling idea generalize to other auxiliary rewards?
2. Could λ be learned dynamically during training (e.g., based on question difficulty or model confidence) rather than fixed manually?

---

> ### Author Response · Authors · 2025-11-20
>
> We thank the reviewer for providing constructive comments. Below we would like to answer the raised questions.
>
> **Q1:** Evaluation focuses only on math reasoning; unclear applicability to other domains.
>
> **A:** To evaluate DRPO’s performance on **non-mathematical reasoning** tasks, we conducted further experiments on the logic puzzle reasoning task. Using K&K logic puzzle dataset [r1],  we train1.5B models for all methods for 400 steps and conduct evaluation every 80 steps. The results are summarized in the following Table and Figure 5 in the revised manuscript. We observed that DRPO method still exhibits a substantially better trade-off than all other baselines, demonstrating the generalizability of the proposed method to other tasks. Notably, DRPO effectively reduces the generation length from 2095 to 1400 (33.2\% length reduction) without any performance drop (from an accuracy of 0.972 to 0.974 ).
>
> |                                            | pass@1/length | pass@1/length | pass@1/length | pass@1/length |
> | ------------------------------------------ | ------------- | ------------- | ------------- | ------------- |
> | DRPO  |  0.972/2095 ($\lambda = +\infty$)    |  0.974/1689 ($\lambda = 0.2$)    |  0.978/1542   ($\lambda = 0.1$)  |  0.974/1400 ($\lambda = 0.07$) |
> | RLOO-LP          |  0.875/2666 ($\alpha=0$)   |  0.868/2404  ($\alpha=0.1$)  |  0.851/1915 ($\alpha=0.2$)   |  0.818/1785 ($\alpha=0.4$)   |
> | ALP         | 0.833/3297  ($\beta=0$)  | 0.788/2543  ($\beta=1e^{-9}$)  | 0.821/1754 ($\beta=1e^{-8}$)   | 0.725/1453  ($\beta=1e^{-7}$)  |
> | HAPO        | 0.878/2420 ($w=0$)    |  0.701/2183 ($w=0.01$)   |  0.718/1520  ($w=0.1$)  |  0.636/1078 ($w=1$)   |
>
> **Q2:** The evaluation relies mainly on Pass@1, which can be unstable for reasoning tasks; additional metrics (e.g., Pass@k) would strengthen the empirical validation.
>
> **A:** Thank you for the constructive comment. In our paper, the reported Pass@1 is computed by averaging over 16 sampled responses for each question to ensure reliability. This evaluation protocol is widely used in recent reasoning works, including those of our baseline methods and DeepSeek-R1 [r2]. Pass@1 is also a more stringent metric than Pass@k (for k > 1), as it requires the model to produce a correct answer in one attempt. Nevertheless, we also include pass@16 as a complementary metric as suggested in Appendix A.7. As shown in Figure 7 with pass@16 metric, DRPO method consistently outperforms all other baselines by a large margin, exhibiting a substantially better trade-off and highlighting the robustness of our approach across metrics.
>
> **Q3:** Could the decoupling idea generalize to other auxiliary rewards?
>
> **A:** Yes! The decoupling idea can be generalized to other auxiliary rewards. In DRPO, the only assumption is that the responses can be decomposed into positives and negatives that are optimized under the discriminative framework. Our length reward $r_l(o)$ in Eqn.(5) can be replaced by any other auxiliary rewards or even combined multiple auxiliary rewards. We leave exploring other auxiliary rewards for future work since the focus of this work is training efficient reasoning models.
>
> **Q4:** Could $\lambda$ be learned dynamically during training (e.g., based on question difficulty or model confidence) rather than fixed manually?
>
> **A:** Yes! We explored adjusting $\lambda$ based on question difficulty estimated by the solve rate $p(q)$ of rollouts in training. Specifically, with a 1.5B model,  we set $\lambda$ as $1- p(q)$ for each question $q$, following the intuition that questions with high solve rates (easy questions) need smaller $\lambda$. The results are presented in the table below. While tuning $\lambda$ provides flexibility in trading off accuracy and efficiency, DRPO with $\lambda=1-p(q)$ may achieve better AES score than fixed $\lambda$, showing the potential of incorporating dynamic $\lambda$. Since setting $\lambda$ as $1- p(q)$ is a quite heuristic strategy, more principled and carefully designs may lead to larger improvements, which we leave for future investigation.
>
> | Method                | Pass@1 | Length | AES   |
> | --------------------- | ------ | ------ | ----- |
> | DisCO                 | 0.649  | 3493   | \-    |
> | DRPO($\lambda=0.5$)    | 0.642  | 2793   | 0.093 |
> | DRPO($\lambda=0.2$)    | 0.632  | 2004   | 0.164 |
> | DRPO($\lambda=0.1$)    | 0.624  | **1527**   | 0.178 |
> | DRPO($\lambda=1-p(q)$) | 0.640  | 2370   | **0.183** |
>
>
> [r1] Xie, Chulin, et al. "On memorization of large language models in logical reasoning." 2024.
> [r2] Deepseek AI. "Deepseek-r1: Incentivizing reasoning capability in llms via reinforcement learning." 2025.

---

### Official Review · Reviewer_tZfR · 2025-11-01

**Soundness:** 3
**Presentation:** 2
**Contribution:** 3
**Rating:** 6
**Confidence:** 3

**Summary:**

This paper focuses on the "overthinking" problem in Large Reasoning Models. Existing reinforcement learning-based solutions (e.g., GRPO) generally lead to performance degradation, which the paper attributes to the fact that current methods incorporating length rewards set the advantage value of correct but verbose reasoning to negative due to GRPO's group-relative advantage function, resulting in inverse optimization. To address this issue, the paper proposes the Decoupled Reward Policy Optimization (DRPO) framework, which is core-based on the DisCO discriminative optimization framework. It decouples the learning signals of correct and incorrect reasoning, normalizes length rewards only within the positive sample group, and integrates the optimized positive data distribution that maximizes length rewards under KL regularization, enabling efficient computation using only on-policy data. The paper validates the framework on multiple mathematical reasoning tasks: the 1.5B model achieves a 77% reduction in length with only a 1.1% performance loss on GSM8k, while the 7B model achieves a 51% reduction in length with only a 2.6% performance loss. These results are significantly superior to baseline methods, with a positive Efficiency-Accuracy Score.

**Strengths:**

- The method design is elegant. By decoupling the learning signals of correct and incorrect reasoning and normalizing length rewards only within the group of correct reasoning samples, it ensures that the rewards for correct but verbose reasoning are only proportionally reduced without becoming negative. The derivation of the optimal positive data distribution is particularly elegant.
- The performance is significant. The experiments demonstrate excellent performance across datasets of varying difficulty levels, achieving significant reductions in output length while maintaining nearly unchanged performance, outperforming baselines.

**Weaknesses:**

- The premise for DRPO to work is that positive and negative samples can be optimized separately, which strongly relies on the DISCO algorithm. This somewhat weakens the contributions of this paper.
- Missing references. It is recommended that the authors add discussions on the following works [1-4] in the 'large reasoning models' part of related work.
- The presentation of this paper is not very user-friendly, and some parts are hard to follow.

[1] Lyu C, Gao S, Gu Y, et al. Exploring the limit of outcome reward for learning mathematical reasoning[J]. arXiv preprint arXiv:2502.06781, 2025.

[2] Zheng C, Liu S, Li M, et al. Group sequence policy optimization[J]. arXiv preprint arXiv:2507.18071, 2025.

[3] Chen A, Li A, Gong B, et al. MiniMax-M1: Scaling Test-Time Compute Efficiently with Lightning Attention[J]. arXiv preprint arXiv:2506.13585, 2025.

[4] Bai L, Cai Z, Cao Y, et al. Intern-s1: A scientific multimodal foundation model[J]. arXiv preprint arXiv:2508.15763, 2025.

**Questions:**

- Does the author consider some operations on the length optimization for the negative sample? At present, there is only the part for the positive sample in the design of the paper.

---

> ### Author Response · Authors · 2025-11-20
>
> We thank the reviewer for providing a comprehensive review and acknowledging that our method design is elegant. Below we would like to address raised concerns.
>
> **Q1:** The premise for DRPO to work is that positive and negative samples can be optimized separately, which strongly relies on the DISCO algorithm. This somewhat weakens the contributions of this paper.
>
> **A:** While DRPO and DisCO both optimize positive and negative samples separately, there exist key differences between them.
>
> (1) DRPO is motivated by our identifying the deficiency of GRPO-based efficient reasoning methods, while DisCO was proposed to achieve good reasoning performance.
>
> (2) DRPO overcomes the key deficiency of GRPO-based efficient reasoning methods by using non-binary length rewards. In contrast, DISCO can handle only binary rewards, leaving its reasoning length uncontrolled, as illustrated in Figures 2 and 3.
>
> (3) DRPO’s objective is rigorously derived from a length-reward–maximizing positive data distribution, yet it can be optimized efficiently using only on-policy data, without introducing additional computational overhead.
>
> Therefore, DRPO is not a trivial extension of DISCO since it exhibits genuine merits in algorithm design to address different challenges in reasoning models.
>
> **Q2:** Adding more references.
>
> **A:** Thank you for pointing out these important works. We've incorporated these works in our revision. Please refer to the revised related work.
>
> **Q3:** About the presentation of this paper.
>
> **A:** Thank you for the feedback. We would appreciate any specific suggestions on which sections or parts you found unclear and will improve them in the revision..
>
> **Q4:** Does the author consider some operations on the length optimization for the negative sample?
>
> **A:** Regarding negative samples, we intentionally leave them unmodified to encourage the model to prioritize finding a correct solution regardless of response length. As negative samples should be penalized to improve accuracy, naively restricting the length of negative samples may limit the model’s ability to explore valid reasoning paths.

---

### Meta-Review · Area_Chair_2JzE · 2025-12-15

**Summary:**

Reviewer tZfR primarily questioned the technical novelty compared to prior work and noted issues with writing clarity. Reviewer ziRG raised concerns about the model's performance in general domains beyond mathematics and suggested additional evaluation metrics. Reviewer Z7wp questioned the rationale behind the proposed method, noted missing baselines (specifically vanilla GRPO), and pointed out potential flaws in using token counts as a proxy for efficiency. Reviewer F8KC noted the lack of experiments on LLMs exceeding 30B parameters and the absence of non-mathematical reasoning tasks and ablation studies.

**Reviewer Concerns:**

Concerns raised by reviewer tZfR are relatively minor and are mainly about a few missing references. The authors have addressed these concerns in their revisions.

Concerns raised by Reviewer tZfR were minor, mainly regarding missing references, which the authors have addressed in the revision. For Reviewer ziRG, the authors provided new experiments on logic puzzles and included Pass@16 results in the appendix, successfully addressing the reviewer's concerns. For Reviewer Z7wp, the authors clarified their motivation and provided comparative results against GRPO that support their claims. While the AC acknowledges that using token count to measure reasoning efficiency is a relatively simple metric, it remains meaningful in this context and does not undermine the technical contribution. For Reviewer Wot1, the inclusion of non-mathematical tasks and reward-length ablations has addressed the primary concerns.

**Reviewer Scores:**

Reviewer tZfR is likely to maintain or increase their positive score. Reviewer ziRG is expected to maintain the high initial rating. Given the compelling quantitative results and clarifications on generalization, the AC think that Reviewers Z7wp and Wot1 would likely to raise to a positive rating if they had participated in the final discussion.

---

### Decision · Program_Chairs · 2026-01-26

Accept (Poster)